# Experimental budgets of OH, HO$_2$ and RO$_2$ radicals and implications for ozone formation in the Pearl River Delta in China 2014

Zhaofeng Tan[1,2,3], Keding Lu[1,3], Andreas Hofzumahaus[2,3,*], Hendrik Fuchs[2,3], Birger Bohn[2,3], Frank Holland[2,3], Yuhan Liu[1], Franz Rohrer[2,3], Min Shao[1], Kang Sun[1], Yusheng Wu[1], LiminZeng[1,3], Yinsong Zhang[1], Qi Zou[1], Astrid Kiendler-Scharr[2,3], Andreas Wahner[2,3], and Yuanhang Zhang[1,3,4,5*]

[1]College of Environmental Sciences and Engineering, Peking University, Beijing, China
[2]Institute of Energy and Climate Research, IEK-8: Troposphere, Forschungszentrum Juelich GmbH, Juelich, Germany
[3] International Joint laboratory for Regional pollution Control (IJRC)
[4] Beijing Innovation Center for Engineering Sciences and Advanced Technology, Peking University, 100871, Beijing, China
[5] CAS Center for Excellence in Regional Atmospheric Environment, Chinese Academy of Sciences, Xiamen, China

*Correspondence to:* Andreas Hofzumahaus (a.hofzumahaus@fz-juelich.de), and Yuanhang Zhang (yhzhang@pku.edu.cn)

**Abstract.** Hydroxyl (OH) and peroxy radicals (HO$_2$, RO$_2$) were measured in the Pearl River Delta which is one of the most polluted areas in China, in autumn 2014. The radical observations were complemented by measurements of OH reactivity (inverse OH lifetime) and a comprehensive set of trace gases including carbon monoxide (CO), nitrogen oxides (NOx = NO, NO$_2$) and volatile organic compounds (VOCs). OH reactivity was in the range between 15 s$^{-1}$ and 80 s$^{-1}$, of which about 50% was unexplained by the measured OH reactants. In the three weeks of the campaign, maximum median radical concentrations were $4.5\times10^6$ cm$^{-3}$ for OH at noon, and $3\times10^8$ cm$^{-3}$ and $2.0\times10^8$ cm$^{-3}$ for HO$_2$ and RO$_2$, respectively, in the early afternoon. The completeness of the daytime radical measurements made it possible to carry out experimental budget analyses for all radicals (OH, HO$_2$, and RO$_2$) and their sum (ROx). The maximum loss rates for OH, HO$_2$, and RO$_2$ reached values between 10 ppbv/h and 15 ppbv/h during daytime. The largest fraction of this can be attributed to radical interconversion reactions while the real loss rate of ROx remained below 3 ppbv/h. Within experimental uncertainties, the destruction rates of HO$_2$ and the sum of OH, HO$_2$, and RO$_2$ are balanced by their respective production rates. In case of RO$_2$, the budget could be closed by attributing the missing OH reactivity to unmeasured VOCs. Thus, the presumption of the existence of unmeasured VOCs is supported by RO$_2$ measurements. Although the closure of the RO$_2$ budget is greatly improved by the additional unmeasured VOCs, a significant imbalance in the afternoon remains indicating a missing RO$_2$ sink. In case of OH, the destruction in the morning is compensated by the quantified OH sources from photolysis (HONO, O$_3$), ozonolysis of alkenes and OH recycling (HO$_2$+NO). In the afternoon, however, the OH budget indicates a missing OH source of (4-6) ppbv/h. The diurnal variation of the missing OH source shows a similar pattern as that of the missing RO$_2$ sink so that both largely compensate each other in the ROx budget. These observations suggest the existence of a chemical mechanism that converts RO$_2$ to OH without the involvement of NO, increasing the RO$_2$ loss rate at daytime from 5.3 ppbv/h to 7.4 ppbv/h, on average. The photochemical net ozone production rate calculated from the reaction of HO$_2$ and RO$_2$ with NO yields a daily integrated amount of 102 ppbv ozone with daily integrated ROx primary sources being 22 ppbv in this campaign. The produced ozone can be attributed to the oxidation of measured (18%) and unmeasured (60%) hydrocarbons, formaldehyde (14%) and CO (8%). An even larger integrated net ozone production of 140 ppbv would be calculated from the oxidation rate of VOCs with OH, if HO$_2$ and all RO$_2$ radicals would react with NO. However, the unknown RO$_2$ loss (evident in the RO$_2$ budget) causes 30 ppbv less ozone production than would be expected from the VOC oxidation rate.

# 1 Introduction

Hydroxyl radicals (OH) constitute the major atmospheric oxidant which is produced in the global troposphere by UV photolysis of ozone (Levy, 1971). The photolysis produces electronically excited $O(^1D)$ atoms which react with water molecules to form OH. In the polluted lower atmosphere, OH can also be produced efficiently by photolysis of nitrous acid (HONO). The reaction with OH initiates the degradation of most trace gases (e.g., CO, VOCs), which leads in many cases to the formation of peroxy radicals, including hydroperoxy radicals ($HO_2$) and organic peroxy radicals ($RO_2$, R = alkyl group). In the presence of nitric oxide (NO), peroxy radicals recycle OH, an important mechanism that increases the oxidizing power of the atmosphere. The above reactions and other fundamental reactions controlling the formation and destruction of ROx radicals are listed in Table 1 (Ehhalt, 1999). The reaction of peroxy radicals with NO has another important implication. $NO_2$ is formed as a product which can be photolyzed. The photodissociation produces ground-state oxygen atoms $O(^3P)$, which combine with $O_2$ and form ozone. The combination of these reactions (R8, R9, and R11) establishes the basic mechanism of photochemical ozone formation in the troposphere (Fishman et al., 1979).

China is a country with a large population and a fast-growing economy, which has caused increasing air pollution during the last decade (Chan and Yao, 2008). However, only few field campaigns have been carried out so far studying photochemistry under polluted conditions in China with the support of OH and peroxy radical measurements. The campaigns were performed in the Pearl River Delta in southern China (Lu et al., 2012; Hofzumahaus et al., 2009) and in the North China Plain around Beijing (Tan et al., 2018b; Tan et al., 2017; Fuchs et al., 2017; Lu et al., 2013; Tan et al., 2018a). Both large regions are densely populated and characterized by air pollution from energy production, traffic, industry, and farming. Chemical box model simulations of OH concentrations have shown general good agreement with measured OH in these regions at NO concentrations above 1 ppbv, but a tendency to underpredict the measured OH at less than 1 ppbv NO. The largest underprediction of OH by a factor of 3 - 5 was observed in Backgarden near Guangzhou in Pearl River Delta (PRD) in summer 2006 (Hofzumahaus et al., 2009). The PRIDE-PRD 2006 (Program of Regional Integrated Experiments of Air Quality over the Pearl River Delta, 2006) campaign was characterized by high OH reactivities with mean daytime values of (20 - 50) s$^{-1}$, where biogenic isoprene and its oxidation products made a contribution of 40% in the afternoon. The results of this campaign showed similarities to those from studies in forested regions, where a model underprediction of OH by up to a factor of ten was reported in isoprene rich air (Tan et al., 2001; Ren et al., 2008; Lelieveld et al., 2008; Pugh et al., 2010; Whalley et al., 2011). The unexplained OH in forests and in the Pearl River Delta was generally assumed to be caused by an unknown OH recycling mechanism that is likely linked to the photochemical degradation of isoprene. In fact, subsequent research discovered previously unknown HOx regeneration reactions, which involve unimolecular isomerisation and decomposition reactions of $RO_2$ in the oxidation of isoprene (Peeters et al., 2014; Peeters and Muller, 2010; Peeters et al., 2009; da Silva, 2010; Crounse et al., 2011; Fuchs et al., 2013; Teng et al., 2017) and methacrolein (Crounse et al., 2012; Fuchs et al., 2014). These reactions do not require NO and explain an OH enhancement by a factor of about two when the OH loss is dominated by isoprene (Fuchs et al., 2013). The mechanism is, however, not sufficient to resolve the large model-measurement discrepancies previously reported.

Another possible reason for the high OH observations could be an interference in the OH measurements. Artefacts of up to 80% have been reported when OH was measured by laser-induced fluorescence (LIF) in forest environments (Mao et al., 2012; Hens et al., 2014; Novelli et al., 2014; Feiner et al., 2016). The interference was presumably caused by oxidation products of biogenic VOCs in the measuring system and was quantified by chemical modulation of ambient OH before the measured air enters the device. When the measurements were corrected for the artefact, the measured OH could be explained by a photochemical box model (Hens et al., 2014; Feiner et al., 2016). In other field studies, where different LIF instruments were used, chemical modulation resulted in only marginal unexplained OH signals at the instrumental limit of detection. One study was in an isoprene

rich forest (Griffith et al., 2016), the other in the polluted North China Plain (Tan et al., 2017). In both studies, the models could explain the OH measurements relatively well, but showed a tendency to underpredict OH under low-NOx conditions (NO < 300 pptv). These discrepancies could not be explained by OH measurement artefacts. Thus, while some of the previously reported high OH observations could have been affected by measurement interferences, there are also indications for an incomplete understanding of radical recycling in VOC rich atmosphere.

In the present paper, we report new radical measurements in PRD during the campaign PRIDE-PRD2014 together with measurements of a large set of atmospheric trace gases. The campaign took place in a suburban area near Guangzhou and was carried out in autumn 2014, the enhanced pollution period for PRD, to elucidate the radical chemistry and secondary pollution formation. Compared to the previous campaign PRIDE-PRD2006, $RO_2$ was measured in addition to OH and $HO_2$ radicals. Chemical modulation was applied to test the OH instrument with respect to possible OH interferences. In the present work, an experimental budget analysis approach is used to quantify the main chemical reactions that control the radical abundances. This approach is possible, as the production and destruction rates of all radical species can be constrained by measurements. The concept has been applied before for OH, when its loss rate could be constrained by OH reactivity measurements (e.g., Shirley et al. ,2006; Hofzumahaus et al., 2009; Whalley et al., 2011; Griffith et al., 2013; Hens et al., 2014). However, the budget analysis for $HO_2$ and $RO_2$ was generally done by chemical box models due to missing $RO_2$ measurements. Here, the completeness of the radical measurements at daytime is used for the first time to quantify the production and destruction processes for all radicals (OH, $HO_2$, $RO_2$) and their sum (ROx) and to analyze their role in photochemical ozone formation. New evidence is found for missing radical recycling under low NOx conditions which is a source of OH, but not of ozone.

## 2 Methodology

### 2.1 Measurement site

The field campaign took place at a long-term monitoring station, the Guangdong Atmospheric Supersite of China (112.929° E, 22.728° N), which is located about 6 km south-west of the city of Heshan. The site is situated on a 60 m high hill, surrounded by woods, small villages, and factories in the surrounding area. A closest highway is about 2 km away to the North West and showed moderate traffic during the measurement campaign. Two major cities, Guangzhou and Foshan, are located at a distance of 50 km north-east of the site. The measurement site is located about 140 km south west of the measurement site (Backgarden) of the former PRIDE-PRD 2006 campaign. The campaign was carried from October to November to be representative for the photochemical polluted season in PRD (Zhang et al., 2008).

### 2.2 Instrumentation

An overview of the instrumentation used for the radical budget analysis is given in Table S1. The inlets of all instruments were located close to the radical measurement inlets at a height of 1.7 m above the roof of the building and 20 m above ground.

#### 2.2.1 Radicals

OH and $HO_2$ radicals were measured by the Peking-University Laser Induced Fluorescence system (PKU-LIF), which was built by Forschungszentrum Jülich (FZJ). OH radicals are detected by laser-induced fluorescence at 308 nm in a gas expansion inside a low-pressure (4 hPa) measurement cell (Hofzumahaus et al., 1996;Holland et al., 2003). In a second cell, $HO_2$ radicals are chemically converted with NO yielding OH, which is detected by LIF (Fuchs et al., 2011). The PKU-LIF instrument was applied first at Wangdu in the North-China Plain in summer 2014 (Tan et al., 2017). It was then moved to the Heshan site for the present

study. At both sites, the PKU-LIF system was complemented by devices from FZJ to measure $RO_2$ (ROx-LIF) and OH reactivity ($k_{OH}$). In the ROx-LIF system, the radicals $RO_2$, $HO_2$, and OH are quantitatively converted to $HO_2$ in a pre-reactor by addition of 0.7 ppmv of NO and 0.11% of CO at a total pressure of 25 hPa. In a second stage at lower pressure (4 hPa), the $HO_2$ is further converted by a large excess of 0.5% NO to OH, which is then detected by LIF (Fuchs et al., 2008). $RO_2$ concentrations are calculated from the total sum of ROx (from ROxLIF) by subtracting the contributions of OH and $HO_2$ measured in the other two detection chambers. A detailed description of the whole measurement system is given by Tan et al. (2017). Due to a technical problem, the integration time for the radical measurements in Heshan was increased to 5 min to achieve $1\sigma$ detection limits of $3.9\times10^5 cm^{-3}$ for OH, $1.2\times10^7 cm^{-3}$ for $HO_2$ and $0.6\times10^7 cm^{-3}$ for $RO_2$ (Table S1). The accuracy of the calibrations depends on the uncertainty of the calibration source (10%, $1\sigma$) and the reproducibility of the calibrations, resulting in total accuracies of ±13%, ±20% and ±26% for OH, $HO_2$, and $RO_2$, respectively (Table S1).

As outlined by Tan et al. (2017), attention was paid to possible interferences in the measurement of HOx. Like in the Wangdu campaign, chemical modulation was applied on several occasions to test whether the OH measurements obtained normally by laser wavelength modulation are perturbed by artificial OH formation in the detection cell. Such kind of interference has been detected in some LIF instruments by applying chemical modulation in field campaigns (Mao et al., 2012; Novelli et al., 2014; Feiner et al., 2016). The chemical modulation system used at Wangdu and Heshan consisted of a flow tube in front of the OH measurement cell and allowed to scavenge ambient OH by addition of propane in the sampled air flow. Switching between propane and nitrogen additions allows discriminating ambient OH from instrumental OH. For nitrogen, we used research grade purity (>99.9990%). A GC analysis of the nitrogen showed no significant contamination by VOCs, which would scavenge OH in the chemical modulation system. In the OH detection cell, scavenging of artificially produced OH by the added propane is calculated to be less than 0.3%. A description of the prototype chemical-modulation reactor used with PKU-LIF is given by Tan et al. (2017). Results of the chemical modulation experiments during the PRIDE-PRD2014 campaign are presented in Section 3.3.

The detection of $HO_2$ by chemical conversion with NO has two known interferences. First, particular $RO_2$ radicals (called $RO_2^{\#}$) from large alkanes (> C4), alkenes (including isoprene) and aromatics can be converted to OH in the $HO_2$ detection cell, leading to a systematic positive bias of the $HO_2$ measurement (Nehr et al., 2011;Fuchs et al., 2011;Whalley et al., 2013;M. Lew et al., 2018). The interference is most effective when the amount of added NO is sufficiently high to convert most of the atmospheric $HO_2$ to OH in the LIF cell. In the campaigns in Wangdu and Heshan, the concentration of the NO reagent was lowered by more than a factor of ten, thereby reducing the interference in the $HO_2$ cell to less than 5% (Tan et al., 2017).

On the other hand, the chemical conversion of $RO_2^{\#}$ can be used intentionally for $RO_2^{\#}$ concentration measurements (Whalley et al., 2013). For that purpose, ROxLIF was operated in an additional measurement mode where the NO addition in the pre-reactor was temporarily turned off and replaced by nitrogen ($N_2$) (Tan et al., 2017). In this mode, the sum of $HO_2$ and $RO_2^{\#}$ is detected in the connected LIF cell, which is operated with a large amount of NO for chemical conversion. The concentration of $RO_2^{\#}$ is then determined as the difference of $HO_2+RO_2^{\#}$ (ROxLIF) and $HO_2$ ($HO_2$ cell). The experimental error of the difference is quite large and exceeds occasionally 100% in the present study, because $RO_2^{\#}$ was much smaller than $HO_2$ (cf. Fig. 1). Also, the calibration error of $RO_2^{\#}$ is larger than that of $HO_2$. The detection sensitivity for individual $RO_2^{\#}$ species lies in the range of (0.8 ± 0.2) times the detection sensitivity for $HO_2$ (Fuchs et al., 2011;Lu et al., 2012). As the speciation of atmospheric $RO_2^{\#}$ is not exactly known, the possible range of sensitivities causes an additional error of 25% that has to be added to the normal calibration error yielding a total accuracy of ±32% ($1\sigma$).

The second NO-related artifact, which is relevant for measurements in the $HO_2$ cell and ROx-LIF system, comes from spurious OH signals generated by the addition of NO in the LIF cells. The signal was regularly determined in humidified synthetic air

during calibration and found to be stable over the campaign. It is equivalent to $(2\pm1)\times10^7\,cm^{-3}$ and $(1\pm1)\times10^7\,cm^{-3}$ for $HO_2$ and $RO_2$, respectively, and is routinely subtracted from the measurements.

The total OH reactivity was measured by an instrument based on laser-flash photolysis laser-induced fluorescence (LP-LIF) (Fuchs et al., 2017;Lou et al., 2010). Ambient air is sampled and pulled through a laminar flow tube at ambient conditions. Artificial OH is produced in the sampled air by pulsed laser photolysis (266 nm, FWHM 10 ns) of ozone, which produces OH in nanoseconds according to reaction R2. The OH decay due to the reaction with atmospheric trace gases is then monitored in real-time by LIF. $k_{OH}$ is determined as a pseudo first-order rate coefficient from the decay curves. The precision of the measured $k_{OH}$ data is $\pm$ 0.3 $s^{-1}$ (1$\sigma$) and the accuracy is 10% (Table S1) at an integration time of 180 s.

### 2.2.2 Trace gases and photolysis frequencies

As summarized in Table S1, the instrumentation used in the PRIDE-PRD2014 campaign was similar to that used in Wangdu (Tan et al., 2017; Fuchs et al., 2017). Photolysis frequencies were determined from spectral actinic photon-flux densities measurements (Bohn et al., 2008). Meteorological parameters, including relative humidity, ambient pressure, and temperature, as well as wind speed and direction, were also regularly measured at the site.

NO and $NO_2$ were measured by a commercial chemiluminescence instrument (Thermo Electron model 42i). $NO_2$ was converted to NO by a custom-built photolytic converter instead of an original molybdenum converter. $O_3$, $SO_2$, CO and $CO_2$ measurements were also measured by commercial instruments from Thermo Electron (model 49i, 43i-TLE, 48i-TLE and 410i). Greenhouse gases, including CO, $CO_2$, $CH_4$, and $H_2O$ were measured by a cavity ring-down spectroscopy instrument (Picarro model G2401). The CO and $CO_2$ measurements from the Thermo Electron and Picarro instruments agreed within the instrumental accuracies. The Picarro measurements were used in this work due to the better data coverage. HONO measurements were performed using a custom-built long-path absorption photometer (LOPAP) from PKU (Liu et al., 2016). Measurements of non-methane hydrocarbons (NMHCs) were performed by a gas chromatograph (GC) using a flame ionization detector (FID) and mass spectrometer (MS) for detection. The GC-FID/MS system provided measurements of $C_2$-$C_6$ alkanes, $C_2$-$C_6$ alkenes, and $C_6$-$C_8$ aromatics (Wang et al., 2014). Formaldehyde was measured by a commercial Hantzsch fluorimeter instrument (Aerolaser GmbH model AL4021). A list of the measured VOCs is given in the Supplement (Table S2).

### 2.3 Experimental radical budget calculations

The radical budget analysis in this work is applied to OH, $HO_2$, $RO_2$, as well as to the whole ROx family. The analysis is based on the chemical mechanism in Table 1, which describes fundamental reactions controlling the abundance of the radicals in the lower troposphere. The reactions include radical chain propagation reactions, which convert one radical species into another one, initiation reactions that produce radicals from closed-shell molecules, and chain termination reactions that destroy radicals. For the budget calculations, measurements for all relevant reactants and photolysis frequencies together with published reaction rate coefficients (Table 1) are used. Unlike in model studies, the analysis does not include model-calculated species.

In the budget analysis, the total production and loss rates of each radical species are calculated and compared to each other. Since all radicals are short-lived (the OH lifetime is less than a second, the lifetime of peroxy radicals is in the order of a minute), their concentrations are expected to be in steady-state with total production and loss rates being balanced. In chemical box models, the balance is always enforced by the numerical solver of the rate equations, even if the chemical mechanism is incorrect. In the experimental budget analysis, however, imbalances are possible and indicate either unknown errors of the experimental input data (concentrations, photolysis frequencies, rate coefficients) or an incorrect chemical mechanism.

The concept of an experimental radical budget analysis has been applied to atmospheric OH in previous studies, where the determination of the total OH loss rate was facilitated by the measurement of $k_{OH}$. OH reactivity measurements avoid the problem that some relevant OH reactants may not be captured by direct measurements.

In the present work, the experimental budget analysis is extended to $HO_2$ and $RO_2$ radicals. In the case of peroxy radicals, no
technique exists for the measurement of their atmospheric total reactivity. However, unlike for OH, the number of known reactant species removing peroxy radicals is relatively small. The main reactants are NO and the peroxy radicals themselves, all of which were measured allowing the total loss rates from the individual reactions to be calculated.

### 2.3.1 ROx budget equations

The ROx budget is entirely controlled by initiation and termination reactions. ROx is primarily produced by photolysis of HONO
(R1), $O_3$ (R2) and HCHO (R3), and ozonolysis of alkenes (R4). The total production rate is calculated as

$$P_{ROx} = j_{HONO} \text{[HONO]} + \phi_{OH} j_{O1D} [O_3] + 2 j_{HCHO-r} \text{[HCHO]} + \Sigma_i \{(\phi^i_{OH} + \phi^i_{HO2} + \phi^i_{RO2}) k^i_4 [\text{alkene}]_i [O_3]\} \qquad \text{(E1)}$$

In case of ozone photolysis, $\phi_{OH}$ is the yield of OH from the reaction of $O(^1D)$ with $H_2O$, which competes with collisional deactivation of $O(^1D)$ with M ($N_2$, $O_2$). In the ozonolysis reactions, the yields $\phi^i_{OH}$, $\phi^i_{HO2}$, and $\phi^i_{RO2}$ are specific for each alkene species $i$. The summation of the radical production from ozonolysis is performed over all measured alkenes. This sum may be not
complete if relevant alkenes were not measured (see discussion in Section 4.2).

The total destruction rate of ROx is given by the reactions of OH with NOx (R12, R13), $RO_2$ with NO (R14), and self-reactions of peroxy radicals (R15 - R17).

$$D_{ROx} = (k_{12}[NO_2] + k_{13}[NO])[OH] + k_{14}[NO][RO_2] + 2(k_{15}[RO_2]^2 + k_{16}[RO_2][HO_2] + k_{17}[HO_2]^2) \qquad \text{(E2)}$$

Since $RO_2$ is measured as a sum of organic peroxy radicals, it is treated as a single species. The generalized rate coefficients are
adopted from MCMv3.3.1 (see footnotes in Table 1). Reaction R14, leading to the formation of organic nitrates, competes with reaction R8 which produces $HO_2$ radicals. The branching ratio between reactions R8 and R14 depends on the carbon chain lengths and structure (Atkinson et al., 1982; Lightfoot et al., 1992). The organic nitrate yield generally increases with carbon number and lies typically between 1% for ethyl $RO_2$ and 35% for $RO_2$ of C8 alkanes. Here, a nitrate yield of 5% is assumed. The impact of larger nitrate yields is discussed in Section 4.

The thermal decomposition of $HO_2NO_2$ into $HO_2$ and $NO_2$ (an initiation reaction) and the back reaction to $HO_2NO_2$ (a termination reaction) are not explicitly considered in the budget equations. The two reactions reach a thermal equilibrium within seconds under the conditions of the campaign and have no net effect on the $RO_x$ balance. Likewise, equilibrium is assumed between thermal decomposition of PAN and its formation by the reaction of acetyl peroxy radicals with $NO_2$. Also, this equilibrium is not explicitly considered in the budget equations E1 and E2.

### 2.3.2 OH budget equations

As explained above, the total OH destruction rate can be directly quantified as the product of the OH concentration and the OH reactivity, both of which were measured during the PRIDE-PRD2014 campaign.

$$D_{OH} = [OH] k_{OH} \qquad \text{(E3)}$$

The total OH production rate is calculated from the primary (R1, R2, R4) and secondary (R9, R10) sources of OH. The primary
sources are treated in the same way as in the ROx budget. The secondary sources include OH recycling from the reaction of $HO_2$ with NO (R9) and $O_3$ (R10).

$$P_{OH} = j_{HONO} \text{[HONO]} + \phi_{OH} j_{O1D} [O_3] + \Sigma_i \{\phi^i_{OH} k^i_4 [\text{alkene}]_i [O_3]\} + (k_9[NO] + k_{10}[O_3])[HO_2] \qquad \text{(E4)}$$

### 2.3.3 $HO_2$ budget equations

The total production rate of $HO_2$ is calculated from primary sources, i.e. photolysis of HCHO (R3) and ozonolysis of alkenes (R4), and secondary sources which involve the conversion of OH and $RO_2$ to $HO_2$. The treatment of the primary production by reactions R3 and R4 is explained in Section 2.3.1. Photolysis of other OVOCs (besides HCHO) could contribute to the $HO_2$ production, but this is not considered here due to the absence of OVOC measurements.

OH to $HO_2$ conversion can occur by reaction of OH with CO, HCHO, $H_2$, and $O_3$. Under the conditions of the campaign, the reaction rates for $H_2$ and $O_3$ were at least two orders of magnitude smaller than those for the reactions with HCHO (R6) and CO (R7). Therefore, only R6 and R7 are considered in the budget analysis. Also, the reaction of $RO_2$ with NO (R8) constitutes an important secondary source of $HO_2$. Reaction R8 competes with the radical termination reaction R14, for which a nitrate yield of 5% is assumed (see Section 2.3.1). Accordingly, an $HO_2$ yield of 95% is taken for reaction R8. The total $HO_2$ production rate is then calculated as

$$P_{HO2} = 2\, j_{HCHO\text{-}r}\, [HCHO] + \Sigma_i\{\phi^i_{HO2}\, k^i_4[alkene]_i[O_3]\} + (k_6[HCHO] + k_7[CO])[OH] + k_8[NO][RO_2] \qquad (E5)$$

$HO_2$ is chemically removed by reaction with NO, $O_3$, $HO_2$, and $RO_2$, all of which were measured in this campaign. It should be noted that the effective rate constant $k_{17}$ for the self-recombination of $HO_2$ has a water vapor dependence which is taken into account in $k_{17}$ (Table 1). The total $HO_2$ destruction rate is then given by

$$D_{HO2} = (k_9[NO] + k_{10}[O_3] + k_{16}[RO_2] + 2\, k_{17}[HO_2])\, [HO_2] \qquad (E6)$$

As explained in Section 2.3.1, the thermal equilibrium between $HO_2+NO_2$ and $HO_2NO_2$ is not explicitly considered in the budget equations.

### 2.3.4 $RO_2$ budget equations

Primary $RO_2$ production is possible by the ozonolysis of alkenes and the photolysis of OVOCs. Owing to the lack of OVOC measurements (except HCHO) in this study, only ozonolysis is considered as a primary source. It is treated as described in Section 2.3.1.

In a broader sense, also reactions of hydrocarbons with $NO_3$ radicals and chlorine atoms can be considered as primary production processes, because they do not consume ROx species. However, neither $NO_3$ nor Cl were measured. $NO_3$ is produced by reaction of $NO_2$ with ozone. It is generally assumed that during the bright hours of the day, $NO_3$ is predominantly destroyed by photolysis and reaction with NO. Recently, Liebmann et al. (2018) reported measurements in a forested environment in southern Germany demonstrating that more than 25% of daytime $NO_3$ was removed by biogenic VOCs. The possible role of $NO_3$ reactions with VOCs at Heshan is discussed in Section 3.

Cl atoms may play a role in the morning. Gaseous $ClNO_2$ can be formed at night by heterogeneous reaction of $N_2O_5$ with $Cl^-$ ions and photolyze quickly after sunrise producing Cl atoms (Osthoff et al., 2008; Tham et al., 2016; Li et al., 2018). This mechanism followed by the reaction of Cl with VOCs made some contribution to the early morning $RO_2$ production in a previous campaign in summertime in the North China Plain (Tan et al., 2017) and will be discussed in Section 4.2.

The main secondary source of $RO_2$ is the reaction of OH with VOCs. As it is generally difficult to measure all reactive organic compounds in the atmosphere, we follow two different approaches to determine the $RO_2$ production from OH reactions. The first approach calculates the $RO_2$ production rate as the sum of the reaction rates of OH with all measured hydrocarbon species, denoted VOC(1). The resulting total $RO_2$ production rate can be considered as a lower limit.

$$P^{(1)}_{RO2} = \Sigma_i\{\phi^i_{RO2}\, k^i_4[alkene]_i[O_3]\} + \Sigma_j\{\, k^j_5[VOC(1)]_j\}[OH] \qquad (E7)$$

Here, the first sum represents the primary production by ozonolysis, the second term the $RO_2$ production rate by OH reactions.

Another approach estimates the total atmospheric amount of organic reactants, here denoted VOC(2), from the measured OH reactivity (e.g., Shirley et al., 2006; Whalley et al., 2016). For this purpose, the reactivity of measured CO, NO, $NO_2$, HCHO, $SO_2$, and $O_3$ is subtracted from the measured $k_{OH}$ to determine the total reactivity of VOCs that can potentially form $RO_2$. This reactivity is called $k_{OH}(VOC(2))$. This approach makes the implicit assumption that the missing OH reactivity found in the present study (see Section 3.2) is caused by unmeasured VOCs. The $RO_2$ production rate is then given by

$$P^{(2)}_{RO2} = \Sigma_i\{\phi^i_{RO2}\, k^i_4[\text{alkene}]_i[O_3]\} + k_{OH}(VOC(2))\,[OH] \qquad (E8)$$

The $RO_2$ destruction is determined by the reaction with NO (R8, R14) and with other peroxy radicals (R15, R16). These reactions and the thermal equilibrium of PAN are treated as in the ROx budget analysis (Section 2.3.1). Accordingly, the total destruction rate of $RO_2$ can be calculated as

$$D_{RO2} = \{(k_8+k_{14})[NO] + (2k_{15}[RO_2] + k_{16}[HO_2])\}[RO_2] \qquad (E9)$$

Equations E7 and E9 can be adapted for the budget analysis of $RO_2^{\#}$ radicals. In this case, the second term in equation E7 contains only OH reactions of VOCs that are known to produce $RO_2^{\#}$. In equation E9, only the concentration of $RO_2$ at the end of the equation has to be replaced by $RO_2^{\#}$ assuming all $RO_2^{\#}$ reacts with $RO_2$.

# 3 Results

## 3.1 Meteorological and chemical conditions

The complete suite of measurements (radicals, trace gases, meteorological parameters) started on 22 October and ended on 14 November. The weather was generally cloudy with temperatures in the 20°C to 30°C range and water vapor volume mixing ratios were around 2 %. Solar UV radiation showed variability due to cloudy weather conditions as can be seen from the photolysis-frequency variations of $j_{O1D}$ and $j_{NO2}$ (Fig. S1). In this work, conditions with $j_{O1D} > 1\times10^{-6}$ s$^{-1}$ are referred to as daytime conditions lasting from 6:00 to 18:00 local time. After 6 November, the weather changed and became rainy with little photochemical activity. Therefore, the current study was restricted to the time period from 22 October to 5 November. During this period, air transportation was dominated by north-easterly and easterly winds. The time dependence of measured trace gas concentrations is shown in Fig. S1 and median values are listed in Table 2. The chemical conditions are characterized by anthropogenic pollution. High concentrations of ozone were observed with daily maxima reaching 100 ppbv on several days. In the morning and afternoon, median $O_3$ values were 16 ppbv and 69 ppbv, respectively. NO mixing ratios reached maximum values of 10 ppbv, and median values were 3.7 ppbv in the morning and 0.4 ppbv in the afternoon. $NO_2$ mixing ratios were 17 ppbv and 9 ppbv in the morning and afternoon, respectively.

## 3.2 OH reactivity

The measured OH reactivity showed variations in the range between 15 s$^{-1}$ and 80 s$^{-1}$ (Fig. S2), with median values of 32 s$^{-1}$ in the morning and 22 s$^{-1}$ in the afternoon (Fig. 1). OH reactivities ($k^{calc}_{OH}$) that were calculated from measured trace gas concentrations, [i] (Table 2) and their OH reaction rate coefficients ($k_{i+OH}$) show a similar temporal behavior as the measured $k_{OH}$, but underestimate its value systematically during the whole campaign (Fig. S2).

$$k^{calc}_{OH} = \Sigma_i\, k_{i+OH}\,[i] \qquad (E10)$$

The comparison of measured and calculated OH reactivities (Fig. 1) indicates that the measured trace gases account for only half of the atmospheric OH reactivity. The contributions of CO, NOx, and measured NMHCs during daytime were about 10%, 14%, and 20%, respectively. Among the measured NMHCs, the groups of alkanes, alkenes (without isoprene) and aromatics had

similar shares of reactivity during the day. In the night and in the morning, alkenes and aromatics were the dominating hydrocarbons, while isoprene made a contribution with up to 6% of the total $k_{OH}$ during daytime. Formaldehyde was the only measured OVOC. It contributed 5% - 8% during the day. The chemical nature of the missing reactivity (about 50%) is not known, but was likely caused by unmeasured VOCs (see Section 3.7).

## 3.3 OH, HO$_2$, and RO$_2$ concentrations

Time series and median diurnal profiles of the measured radical concentrations are shown in Fig. S2 and Fig. 1. As in previous campaigns, the diurnal profile of OH shows a high correlation with $j_{O1D}$, which is a proxy for the solar UV radiation driving much of the primary radical production during summer time (e.g., Rohrer et al., 2014; Ehhalt and Rohrer, 2000). It is noteworthy that the measured OH persisted even after sunset at levels with median values of $0.7 \times 10^6$ cm$^{-3}$ until midnight (Fig. 1). Thereafter, OH concentrations dropped continuously until they reached values below the limit of detection shortly before sunrise.

In order to test, whether the OH measurements at Heshan could have been affected by an unknown interference, chemical modulation experiments were performed between noon and midnight on several days (Table 3). Within instrumental precision, no significant unexplained OH interference was detected. This result applies equally to day and night. Therefore, the OH concentration of $1.8 \times 10^6$ cm$^{-3}$ measured on 30 October after sunset during the chemical modulation test (Table 3) must be considered as real ambient OH. On the other hand, all unaccounted OH signals are slightly positively biased except the last one in Table 3. The mean value ($\pm$ 1$\sigma$) of the unaccounted OH signals is equivalent to an OH concentration of $(0.3 \pm 0.3) \times 10^6$ cm$^{-3}$ which is at the limit of instrumental detection. In the further analysis, the mean plus 1$\sigma$ is assumed to be the upper limit for a possible OH interference.

The peroxy radicals, HO$_2$ and RO$_2$, show qualitatively a similar diurnal behavior as has been reported for other urban environments (e.g., (Holland et al., 2003; Shirley et al., 2006; Kanaya et al., 2007; Emmerson et al., 2007)). The photochemical build-up of peroxy radicals after sunrise is delayed due to their reaction with high NO concentrations in the morning. The peroxy radical concentrations reach a late maximum around 14:30h in the afternoon when NO is decreasing (Fig. 1). Median ozone concentrations reach a maximum of 76 ppbv at the same time as the occurrence of the peroxy radical maxima. The concentration of RO$_2^{\#}$ represents about $(15 \pm 15)\%$ of the total organic peroxy radicals, RO$_2$, during daytime. This percentage is surprisingly small as measured RO$_2^{\#}$ precursors (alkenes, isoprene, aromatics and large alkanes) dominated the reactivity of the measured hydrocarbons (Fig. 1, lowest panel). A possible reason would be unmeasured VOCs which produce RO$_2$, but not RO$_2^{\#}$.

## 3.4 RO$_x$ budget

In the RO$_x$ budget analysis, only radical initiation and termination reactions play a role. Calculated median diurnal profiles for production (E1) and destruction (E2) rates of ROx are shown in Fig. 2. Both rates have maximum values in the order of 3 - 4 ppbv/h at noontime (Fig. 2e and 2f).

The observed daytime production of RO$_x$ (Fig. 2a - 2e) is dominated with 51% by the photolysis of HONO (R1), followed by photolysis of formaldehyde (R3) and ozone (R2) which have contributions of 34% and 15%, respectively. The counteracting reaction forming HONO from OH and NO is comparatively slow (<5%; Fig. 2b, blue line). Therefore, the net OH production from HONO photolysis is still the major primary source of ROx. Ozonolysis of measured alkenes (R4) contributes 7% during daytime and is the only primary source considered here at night.

RO$_x$ radical termination occurs via reactions with NO$_x$ (R12 - R14) and by peroxy radical self-reactions (R15 - R17). In the morning, when the median NOx concentration is about 21 ppbv (Table 2), the reaction of OH with NO$_2$ (R15) is the dominating loss process (69%), followed by the reaction of RO$_2$ with NO (R14, 13% on average) and the reaction of OH with NO (R13, 7%

on average). Radical self-reactions play a negligible role in the morning. They gain importance (12%) in the afternoon, when the median $NO_x$ concentration decreases to 9.5 ppbv. Overall, ROx loss during the PRIDE-PRD2014 campaign was mainly controlled by reactions with NOx during the whole day.

The difference between the total ROx production and destruction rates is shown in Fig. 2g. During daytime, there is an imbalance between production ($P_{ROx}$) and destruction ($D_{ROx}$) rates, which increases from -0.5 ppbv (8:00) to +0.5 ppbv (18:00). The deviations are small (< 15%) compared to the total ROx turnover rate at noon and can be explained by the experimental errors of the data between 8:30 and 17:30. After sunset until 2 hours after midnight, $D_{ROx}$ remains about 0.5 ppbv/h larger than $P_{ROx}$. The deviation may be caused, at least partly, by $RO_2$ formation from reactions of VOCs with $NO_3$ radicals, for which an upper limit of 0.7 ppbv/h can be estimated (see Section 4.2). The deviation can also largely be explained by experimental errors, when the upper limit of a possible OH interference is included in the total experimental uncertainty (Fig. 2g).

## 3.5 OH budget

The median OH production (E4) and destruction (E3) rates show diurnal profiles with noontime maxima of 13 ppbv/h and 16 ppbv/h, respectively (Fig. 3a). The calculated OH production rate is dominated throughout the whole day by the recycling reaction of $HO_2$ with NO, which contributes on average 79% during the day. The most important primary daytime source is the photolysis of HONO. It contributes on average 13% to the total OH production, whereas the photolysis of ozone and ozonolysis of alkenes add 4% each. The OH destruction rate is balanced by the calculated production rate between sunrise and noon, but is considerably larger than the production rate in the afternoon (Fig. 3b). The difference of (4-6) ppbv/h cannot be explained by the combined experimental uncertainties Fig. 3b), even if the upper limit of the potential OH interference is taken into account (Fig. 3b). Since the measured $k_{OH}$ provides a constraint for the total OH loss rate, the imbalance with $P_{OH}$ less than $D_{OH}$ indicates a significant missing OH source in the calculation of $P_{OH}$. Only after sunset, the remaining discrepancy would be explainable by a potential OH artifact.

## 3.6 HO$_2$ budget

The calculated $HO_2$ production (E5) and destruction rates (E6) are in good agreement throughout the whole day. The maxima of the rates are in the order of (12 - 14) ppbv/h shortly before noontime (Fig. 3c). The major sources are the reactions of $RO_2$ with NO (63%) and the reactions of OH with CO (14%) and formaldehyde (13%). Primary production processes (photolysis of formaldehyde, ozonolysis of alkenes) contribute 10% to the total $HO_2$ production. Owing to the relatively high NO concentrations during the campaign, the $HO_2$ loss is dominated with 96% by the reaction with NO. The $HO_2$ budget is closed within the experimental uncertainties (Fig. 3d). The magnitude of unexplained OH signals observed in the chemical modulation experiments has no noticeable influence on the closure of the budget (Fig. 3d).

## 3.7 RO$_2$ budget

Like for $HO_2$, the destruction rate of $RO_2$ (E9) is dominated with 98% by the reaction with NO and reaches a maximum shortly before noontime (Fig. 3e). In this case, the maximum has a value of (10 - 11) ppbv/h. The production rate $P^{(1)}_{RO2}$ calculated by equation E7 from measured hydrocarbons (Fig. 3e) is far from being able to compensate the loss of $RO_2$ radicals. At noontime, the production rate is a factor of 4 - 5 too small. Once the missing OH reactivity is attributed to unmeasured VOCs, the resulting production rate $P^{(2)}_{RO2}$ calculated by equation E8 matches $D_{RO2}$ relatively well.. From sunrise to noon, the budget becomes closed within the experimental uncertainties (Fig. 3f). This result strongly supports the hypothesis that the missing OH reactivity in the morning is caused by unmeasured VOCs.

Assuming that unmeasured VOCs are also responsible for the missing reactivity at other times of the day, an imbalance of (2 - 5) ppbv/h in the $RO_2$ budget is left in the afternoon, where $P^{(2)}_{RO2}$ is greater than $D_{RO2}$ (Fig. 3f). Considering the experimental uncertainties in the budget equations, the difference is significant from noontime to midnight and indicates a missing $RO_2$ sink. Even if the maximum potential OH interference is taken into account, the imbalance remains from the afternoon until 21:00h, while it becomes insignificant later in the night (Fig. 3c).

The $RO_2^{\#}$ radical budget can be treated in a similar way as for $RO_2$ (see Section 2.3.4). The calculated destruction rate of $RO_2^{\#}$ shows a similar diurnal shape as $RO_2$ and is also entirely determined by the reaction with NO (Fig. 3g). Within experimental uncertainty the destruction rate is balanced by the production rate $P^{(1)}_{RO2\#}$ calculated from the measured VOCs known to produce $RO_2^{\#}$ (Fig. 3h). Note that $P^{(1)}_{RO2\#}$ in panel (g) looks almost the same as $P^{(1)}_{RO2}$ in panel (e) as most of the measured VOCs produce $RO_2^{\#}$.

If the missing OH reactivity was caused entirely by $RO_2^{\#}$ precursor VOCs, the production rate of organic peroxy radicals would be up to factor of 5 higher than the $RO_2^{\#}$ loss rate (Fig. 3). This suggests that the missing OH reactivity is caused by chemical species that do not produce $RO_2^{\#}$.

## 4 Discussion

The completeness of the radical measurements allows a budget analysis for all radicals (OH, $HO_2$, $RO_2$) based on experimental data only, without application of a chemical box model, under the assumption that for the production and loss rates all relevant species were measured. The $RO_x$ budget analysis compares whether the radical initiation reactions of $RO_x$ are balanced by the known radical termination reactions; the analysis of the OH, $HO_2$ and $RO_2$ budgets gives insight into the completeness of our understanding of the radical cycling. The interpretation of the budgets below will focus on daytime chemistry (6:00-18:00), as conclusions concerning the nighttime chemistry would be rather limited due to the possible OH interference and the lack of $NO_3$ measurements.

### 4.1 Missing OH reactivity

Of the atmospheric OH reactivity measured at Heshan, approximately 25% is explained by measured inorganic compounds (CO, NOx) and another 25% by measured NMHCs and formaldehyde. The missing reactivity of about 50% indicates a considerable fraction by unmeasured reactants. Similar missing reactivities have been observed also in forests and other urban environments (Williams et al., 2016; Whalley et al., 2016; Ramasamy et al., 2016; Kaiser et al., 2016; Lu et al., 2013; Edwards et al., 2013; Dolgorouky et al., 2012; Lou et al., 2010; Sadanaga et al., 2005). Depending on the local conditions, missing $k_{OH}$ is generally attributed to unmeasured VOCs which have been emitted or produced by atmospheric oxidation. This hypothesis is plausible, since the atmosphere contains thousands of unknown organic species (Goldstein and Galbally, 2007), but the assumption of unmeasured VOCs is generally difficult to prove. In the present work, the existence of unmeasured atmospheric VOCs deduced from missing OH reactivity is independently confirmed by the analysis of the experimental $RO_2$ budget presented in Section 3.7. Only if the missing reactivity is due to VOCs, the discrepancy of up to a factor of 5 between observed $RO_2$ production and destruction rates can be reconciled (Fig. 3e, f). In addition, the budget analysis for $RO_2^{\#}$ provides evidence that the unmeasured VOCs are mostly species that do not produce $RO_2^{\#}$ radicals (Fig. 3g, h). As such, they probably do not belong to the class of alkenes or aromatics.

Missing reactivity at the measurement site has also been reported by Yang et al. (2017), who analyzed $k_{OH}$ data measured from 20 October to 19 November 2014 using the comparative reactivity method developed by Sinha et al. (2008). Although their time

window encompasses the present study, there is little overlap of the data due to data gaps. As far as simultaneous data are available, the two instruments agreed within their combined errors ($1\sigma = \pm20\%$ for the CRM instrument, $\pm10\%$ for the LP-LIF instrument). For the time period analyzed by Yang et al. (2017) the fraction of missing reactivity to the total reactivity was reported to be 30%. Although the percentage value is smaller than in the present paper (50%), the absolute values for the OH reactivity from unmeasured reactants are comparable. The speciation of the missing reactivity, however, can be different because the higher NOx loading in the period analyzed by Yang et al. (2017) may lead to different photochemical products and may be correlated with different VOC emissions. Missing OH reactivity of about 50% was also reported for the Backgarden site about 140 km north-east of Heshan, where daytime OH reactivities between 20 $s^{-1}$ and 50 $s^{-1}$ were measured in summer 2006 (Lou et al., 2010). In that study, the missing reactivity could be explained by unmeasured OVOCs (e.g., formaldehyde, acetaldehyde, MVK, MACR) which were simulated by a chemical box model as products from measured hydrocarbons. With a similar approach, Yang et al. (2017) explain one to two-thirds of the missing reactivity at Heshan by organic oxidation products (e.g. aldehydes, dicarbonyl compounds) and suspect that the remaining missing reactivity was caused by unknown primary VOC emissions.

Many oxygenated VOCs produce $RO_2$ when they react with OH. Examples are acetaldehyde and higher aldehydes, acetone and higher ketones, MACR, MVK, and methyl glyoxal, all of which were not measured in the present study. In the following discussion we assume based on the considerations given above that the missing reactivity in the present study is entirely due to VOCs (including also OVOCs) which can produce $RO_2$ by reaction with OH.

## 4.2 Radical budgets, their relationships and uncertainties

The imbalances in the OH, $HO_2$, and $RO_2$ budgets (*D-P*) reach median values of up to ($7\pm2.5$) ppbv/h, -($3\pm5$) ppbv/h, and -($5\pm2.5$) ppbv/h, respectively, during the day. Interestingly, the imbalance in the ROx budget does not exceed $\pm0.5$ ppbv/hr (Fig. 2). This means that the largest uncertainties in the speciated radical budgets compensate each other in the ROx budget. The largest differences between destruction and production rates are found for OH (Fig. 3b) and $RO_2$ (Fig. 3f). The respective diurnal profiles look similar in shape, but with opposite signs. When added up in the ROx budget, their values largely compensate each other. The imbalances in the OH and $RO_2$ budgets become large in the afternoon and show a growing trend when NO falls below 1 ppbv (Fig. 4). Above 1 ppbv NO (i.e., in the morning), however, both budgets are closed within their experimental uncertainties. In case of $HO_2$, the destruction and production rates are balanced within experimental error during the whole day independent of the NO mixing ratio (Fig. 3c and Fig. 4).

One possible explanation for the imbalances in the OH and $RO_2$ radical budgets would be an unknown radical initiation reaction for OH and an unknown termination reaction for $RO_2$, respectively, which fortuitously balance each other in time and quantity in the ROx budget. This coincidence seems unlikely, also because this would mean a drastic increase in ROx production and destruction rates by a factor of 2.5 to 3 in the early afternoon. A more plausible explanation is a partially insufficient description of the radical chain propagation, which proceeds during the day at much higher rates of 10 - 16 ppbv/h.

*OH interference*

The non-closure in the OH and $RO_2$ budgets could be explained, for example, by experimental artifacts as recently reported for OH measurements in some LIF instruments (Mao et al., 2012; Novelli et al., 2014; Feiner et al., 2016). An experimental overestimation of the OH concentration would result in too high reaction rates calculated for the OH destruction ($k_{OH}\times$[OH]) and the $RO_2$ production from the reactions of VOCs with OH. However, the chemical modulation tests carried out under low NO conditions (< 1ppbv) do not show a significant interference. This result is consistent with other tests that were performed with

our LIF technique in field campaigns in China (Tan et al., 2018b; Tan et al., 2017) and in laboratory and chamber experiments (Fuchs et al., 2016; Fuchs et al., 2012). For this reason, we consider OH interference as an unlikely explanation here, although we cannot strictly exclude the possibility that there were OH interferences only at times, when the chemical modulation system was not used.

*OH regeneration mechanisms*

An alternative explanation for the non-closure in the OH and $RO_2$ budgets would be a chemical mechanism that effectively converts $RO_2$ to OH. Unimolecular isomerization and decomposition reactions of $RO_2$ can be such an OH source when the competing reaction with NO is slow. This chemistry is known for a long time as autoxidation, for example, in low-temperature combustion (e.g., Cox and Cole, 1985; Glowacki and Pilling, 2010). Its potential relevance for atmospheric chemistry at ambient

temperatures has only recently been recognized (e.g., Peeters et al., 2009; daSilva et al., 2010; Crounse et al., 2013; Praske et al., 2018). Autoxidation involves an intramolecular H-shift in the $RO_2$ molecule leading to a hydroperoxy alkyl radical, which is often named QOOH. This radical can generally undergo various types of reactions such as reaction with $O_2$ producing an oxygenated VOC + $HO_2$, or decomposition to an oxygenated VOC by elimination of OH from the -OOH group (e.g., Peeters et al., 2009; daSilva et al., 2010; Crounse et al., 2013; Praske et al., 2018). Another path is the addition of $O_2$ forming a

hydroperoxy peroxy radical $(O_2)QOOH$. This new peroxy radical can then react with NO, $HO_2$, $RO_2$, or undergo another internal H-shift reaction. Repetitive sequential H-shift reactions followed by $O_2$ addition lead to highly oxidized $RO_2$ radicals which produce highly oxidized molecules (HOMs) by radical termination reactions (e.g., Ehn et al., 2014, 2017; Jokinen et al., 2014). Due to their low vapour pressure, HOMs are efficient precursors for organic particles, which are produced from the original VOCs with yields in the low percent range (e.g., Ehn et al., 2014; Jokinen et al., 2015).

$RO_2$ isomerisation producing HOx radicals is known to occur in the oxidation of isoprene ((Peeters et al., 2014; Peeters and Muller, 2010; Peeters et al., 2009; Da Silva et al., 2010; Crounse et al., 2011; Fuchs et al., 2013; Teng et al., 2017) and methacrolein (Crounse et al., 2012; Fuchs et al., 2014). In Heshan, the production rate of isoprene peroxy radicals from the reaction of isoprene with OH never exceeded 0.5 ppbv/h. Even if every isoprene derived $RO_2$ radical regenerated one OH molecule (which is not likely because of the competing reaction with NO), the process could explain only a small fraction of the

missing OH production rate. The concentration of methacrolein (MACR) was not measured, but is generally not larger than that of isoprene (Karl et al., 2009; Shao et al., 2009). Since the OH rate constant is smaller than that for isoprene, OH regeneration by unimolecular reactions of MACR derived $RO_2$ is expected to be even less important.

Besides for isoprene and methacrolein, autoxidation of $RO_2$ leading to HOx formation has been experimentally studied for only few other VOCs, including 3-pentanone (Crounse et al., 2013), glyoxal (Lockhart et al., 2013), n-hexane and 2-hexanol (Praske

et al., 2018), hydroxymethyl hydroperoxides (Allen et al., 2018), and 2-hydroperoxy-2-methylpentane (Praske et al., 2019). While isoprene and methacrolein chemistry is especially important in biogenically controlled environments, the new studies demonstrate that autoxidation can also be expected to play a role in urban atmospheres when NO concentrations are as low as 500 pptv (cf., Praske et al., 2018). Systematic theoretical studies have shown that the rates of H-shifts in $RO_2$ depend very much on their chemical structure (e.g., Crounse et al., 2013; Otkjær et al., 2018; Møller et al., 2019) and range from $10^{-4}$ $s^{-1}$ to 10 $s^{-1}$ at

ambient temperature. Low rate coefficients can be expected, for example, for 1,5-H or 1,6-H shift reactions in linear alkyl radicals, whereas the presence of (multiple) functional groups like -OH, -OOH, or -CHO may increase the H-shift rate by orders of magnitude. The possible yield of OH, $HO_2$, or higher-oxidized $RO_2$ radicals from a hydrogen shift depends on the chemical structure and functionality. Elimination of OH or $HO_2$ is generally supported by the presence of functional groups (e.g., -OH, -OOH, or -CHO).

In the present campaign, the potential conversion of $RO_2$ to OH by a unimolecular reaction would require a rate of about 0.08 s$^{-1}$ to close the budgets of OH and $RO_2$, if all measured $RO_2$ radicals could produce OH from H-shift reactions (Fig. S4). Although the rate is in the possible range of H-shift reactions, it is questionable if a major fraction of $RO_2$ was structurally capable to undergo a fast H-shift leading to OH formation. As approximately half of the measured OH reactivity is likely due to unmeasured VOCs with unknown speciation and owing to the general lack of kinetic and mechanistic studies for specific $RO_2$ radicals, it is not possible to be more quantitative here. If the missing reactivity at Heshan was caused by photochemically aged, functionalized oxygenated VOCs, there is a chance that $RO_2$ radicals from these compounds could have contributed significantly to the missing $RO_2$ sink, or missing OH source. If autoxidation played a significant role, additional $HO_2$ formation from H-shift reactions would also be expected. However, the closed $HO_2$ budget gives no indication for this.

Another possibility to convert $RO_2$ to OH under low NO conditions is the reaction of $RO_2$ with $HO_2$. The reaction is generally considered to be chain terminating, but in the case of acyl peroxy and $\alpha$-carbonyl peroxy radicals, a parallel reaction channel can produce OH with yields up to 80% (Fuchs et al., 2018; Winiberg et al., 2016; Praske et al., 2015; Groß et al., 2014; Hasson et al., 2012; Dillon and Crowley, 2008; Hasson et al., 2004; Jenkin et al., 2008). With reaction rate constants in the range of $(1 - 2) \times 10^{-11}$ cm$^{-3}$ s$^{-1}$, the OH production would be less than 0.1 ppbv/h even if all measured $RO_2$ species produced OH. In conclusion, the already known mechanisms for conversion of $RO_2$ to $HO_2$ are not sufficient to explain the missing $RO_2$ sink and missing OH source.

The observations in Heshan resemble qualitatively that of the previous PRIDE-PRD 2006 campaign in Backgarden (Hofzumahaus et al., 2009), where the experimental OH budget indicated a missing OH source (28 ppbv/h) in the afternoon, when NO was less than 1 ppbv. Box model simulations underestimated the measured OH concentrations by a factor 3-5 at low NO, but agreed well with measured $HO_2$ concentrations. In that study, the isoprene concentrations were considerably higher reaching several ppbv. Isomerisation reactions of isoprene peroxy radicals could explain only a small part (max. 20%) of the missing OH source (Lu et al., 2012; Fuchs et al., 2013). The observed behavior of OH and $HO_2$ could be reproduced by assuming a hypothetical mechanism in which $RO_2$ is converted to $HO_2$ and $HO_2$ to OH by an unknown reactant X with a concentration equivalent to 0.8 ppbv NO. Owing to the lack of $RO_2$ measurements, the mechanism could not be directly tested for $RO_2$. In Heshan, the same mechanism would be able to close both the $RO_2$ and OH budgets, if an equivalent of 0.4 ppbv NO is assumed (Fig. S4). Here, the NO concentration of 0.4 ppbv corresponds to the rate coefficient of 0.08 s$^{-1}$ discussed above. When using the X mechanism, the closure of the $HO_2$ and $RO_x$ budgets remains unaffected.. Therefore, the X mechanism is one possibility to describe all radical budgets at Heshan consistently, although its chemical nature remains unresolved.

*Additional uncertainties in the budget analyses*

As pointed out in Section 4.1, unmeasured VOCs were most likely responsible for the observed missing OH reactivity. This not only considerably influences the radical chain propagation from OH to $RO_2$ (Fig. 3e, g), but can also affect the primary production of radicals by ozonolysis and photolysis (see Supplementary Text). Furthermore, Cl reactions with measured and unmeasured VOCs may have initiated ROx chain reactions in the early morning. These reactions could have slightly influenced the ROx budget, but are of minor importance compared to the radical chain propagation reactions (Supplementary Text).

Reactions of $NO_3$ with VOCs are an additional $RO_2$ source which is neglected in the budget calculation in Section 2.3.4. The relevance can be estimated from the production rate of $NO_3$, which is calculated from the reaction of $NO_2$ with $O_3$ ($k(NO_2+O_3)=$ $1.47 \times 10^{-13} \times exp(-2470/T)$; MCM3.3.1). In this campaign, the $NO_3$ production rate was in the order of 1.4 ppbv/h and 0.7 ppbv/h at day- and nighttime, respectively. Because $NO_3$ is efficiently photolysed in the bright hours of the day, it is generally assumed that it plays a negligible role as an oxidant during daytime. Liebmann et al. (2018) have recently shown that this is not always the

case. They reported measurements in a forested environment in southern Germany demonstrating that more than 25% of the daytime $NO_3$ reacted with biogenic VOCs. Under the conditions at Heshan 2014, the main loss process at daytime is the reaction with NO. If we neglect unmeasured VOCs, the percentage removal of $NO_3$ in the morning is 96% by NO, 3% by photolysis, and 1% by measured VOCs. In the afternoon, the corresponding values are 72%, 21%, and 7%. Thus, the estimated $RO_2$ production rate from $NO_3$ reactions with known VOCs was probably not more than 0.1 ppbv/h at daytime. It is conceivable, that unmeasured VOCs, which probably accounted for 50% of the OH reactivity, contributed by a similar magnitude. During daytime, these contributions are relatively small compared to the total production rate of $RO_2$. The tendency is to slightly increase the imbalance between the production and destruction rate of $RO_2$ observed in the afternoon (Fig. 3). At sunset and in the night, the $NO_3$ production rate of 0.7 ppbv/h can be considered as an upper limit for the $RO_2$ production. This value can possibly explain at least partly the imbalance of about 0.5 ppbv/h in the ROx budget after sunset (Fig. 2).

Another uncertainty is caused by the measurement and incomplete representation of the $RO_2$ chemistry. Due to the measurement principle of the ROxLIF instrument, only those $RO_2$ species are measured which are converted in the instrument to $HO_2$ by reaction with NO. This measurement is suitable to quantify the $HO_2$ production rate (Equation E5). Among the $RO_2$ radicals which are not completely captured by ROxLIF are those species which produce a new $RO_2$ radical when they react with NO. As these reactions are neutral with respect to the total $RO_2$, the $RO_2$ budget ($D$-$P$) is not sensitive to the $RO_2$ measurement bias caused by these species (see Supplementary Text).

Other uncertainties in the $RO_2$ budget are caused by the rate constants for the reactions of $RO_2$ with NO (Reaction R8, R14), $RO_2$ (Reaction R15), and $HO_2$ (Reaction R16) that are given in Table 1 as effective values for the lumped $RO_2$ radicals. In this work, the uncertainty of the rate coefficients for Reaction R15 and R16 plays only a minor role, because the daytime loss of the peroxy radicals was largely dominated by the reaction with NO (see Supplementary Text). The relevant range for the reaction rate constants of different $RO_2$ species with NO (Reaction R8, R14) is between $8\times10^{-12}$ $cm^3s^{-1}$ and $1.1\times10^{-11}$ $cm^3s^{-1}$ (see Supplementary Text). As a sensitivity test, Figs. S5 and S6 show the budgets of ROx, $RO_2$ and $HO_2$ for a rate constant of $1\times10^{-11}$ $cm^3s^{-1}$. The results are essentially the same as in Figs. 2, 3 where a rate constant of $9\times10^{-12}$ $cm^3s^{-1}$ was applied for Reaction R8 + R14. Thus, an increased rate constant cannot explain, the missing $RO_2$ sink in the $RO_2$ budget.

Also the unknown branching ratio in the reaction of $RO_2$ with NO, which can produce $HO_2$ (R8, chain propagating) or organic nitrates (R14, chain terminating) is uncertain (see Section 2.3.1 and Supplement). Changing the yields for organic nitrates from 5% to 20% has a small, but notable influence on the ROx budget, reversing the slightly negative bias ($D < P$) to a lightly positive one ($D > P$) (Fig. S7). In both limits, the ROx budget remains closed within experimental errors. The influence of the different $HO_2$ yields on the production rate of $HO_2$ is small (Fig. S8). For the range of tested yields, the $HO_2$ budget remains balanced within experimental uncertainties.

**4.3 Photochemical ozone production**

Photochemical ozone production in the troposphere is due to the oxidation of NO to $NO_2$ by reaction with peroxy radicals (R8, R9), followed by $NO_2$ photolysis yielding NO and $O(^3P)$ atoms. The O atoms combine with $O_2$ and form ozone. The net ozone production can be estimated from the production rate of $NO_2$ via reactions R8+R9, corrected for chemical loss of $NO_2$ by reaction with OH (R12) (e.g., Kanaya et al., 2007; Ren et al., 2013; Brune et al., 2016).

$$P^{(1)}{}_{O3} = k_8[RO_2][NO] + k_9 [HO_2] [NO] - k_{12}[NO_2][OH] \qquad (E11)$$

Chemical loss of ozone by photolysis (R2), ozonolysis reactions (R4) and dry deposition is neligible under the given conditions. Calculated losses of ozone by photolysis and ozonolysis are not larger than 0.2 ppbv/h. The dry deposition rate at daytime is estimated to be no more than 1 ppbv/h assuming a mixed boundary layer height of 1km and a maximum deposition velocity of 1

cm/s (e.g., Weseley et al., 2000). Using the rates shown in Fig. 2f (OH+NO$_2$), Fig 3c ($D_{HO2}$) and Fig 3e ($D_{RO2}$), a daily integrated net ozone production of (102±31) ppbv is calculated (06:00 h to 18:00h). For comparison, the daily integrated OH+NO$_2$ term is (14±3) ppbv. About 70% of the ozone is produced in the morning (06:00 - 12:00h) and 30% in the afternoon (12:00 - 18:00h).

The radical budgets for OH, HO$_2$, and RO$_2$ allow tracing back which chemical processes are driving the production of peroxy radicals and therefore ozone formation. The first term ($k_8$[RO$_2$][NO]) in equation E11 can be considered as the contribution of the VOCs that form RO$_2$ which continue to react with NO to HO$_2$. As the HO$_2$ budget is essentially balanced, the HO$_2$ loss term ($k_9$ [HO$_2$] [NO]) can be replaced by the rate of HO$_2$ producing processes (R3, R6, R7, R8) shown in Fig. 3c. This replacement implicitly assumes that other HO$_2$ losses such as HO$_2$+RO$_2$ and HO$_2$+HO$_2$ are negligible, which is valid during this study. The ozone production from RO$_2$ and HO$_2$ can then be expressed as

$$P^{(2)}_{O3} \cong 2 \times k_8[RO_2][NO] + 2 \times j_{HCHO-r} [HCHO] + (k_6[HCHO] + k_7[CO])[OH] - k_{12}[NO_2][OH] \tag{E12}$$

Using equation E12 yields a daily net-ozone production of 112 ppbv (Fig. 5). This value is in close agreement with the result of equation E11 which is using different experimental input parameters. According to equation E12, a percentage of 78% of the daily net-ozone production results from the oxidation of VOCs via reactions R5, R8, and R9, 14% from reactions of HCHO (R3+R6, followed by R9) and 8% from CO oxidation (R7, R9). Measured VOCs that produce RO$_2$ account for only 18% of the total ozone production, while unmeasured VOCs contribute the dominant fraction of 60%.

In principle, the first term ($k_8$[RO$_2$][NO]) in equations E11 or E12 could be replaced by the production rate calculated from the total VOC reactivity ($k_{OH}$(VOC(2))×[OH]), if the RO$_2$ budget was balanced.

$$P^{(3)}_{O3} \cong 2 \times k_{OH}(VOC(2)) \times [OH] + 2 \times j_{HCHO-r} [HCHO] + (k_6[HCHO] + k_7[CO])[OH] - k_{12}[NO_2][OH] \tag{E13}$$

Using this equation, an integrated net-ozone production of 140 ppbv would be calculated. However, the RO$_2$ budget is not balanced (Fig. 3e) and indicates a missing RO$_2$ sink, which does not oxidize NO, and therefore does not produce ozone. This possibility was first suggested when the RO$_2$ to OH conversion by X was proposed to explain a missing OH source in PRD 2006 (Hofzumahaus et al., 2009). In the present case, such an RO$_2$ sink would remove 15 ppbv (22%) of the daily produced RO$_2$ resulting in an integrated ozone production that is 30 ppbv lower than expected from the rate of VOC oxidation.

The possible underprediction of the photochemical ozone production from unknown (unmeasured) atmospheric VOCs has been pointed out in previous studies, where RO$_2$ concentrations have been modelled (e.g., Griffith et al., 2016) or estimated from OH reactivities (Whalley et al., 2016). In the ClearfLo campaign 2012 in central London, the ozone production calculated from the oxidation of C2-C8 VOC species (measured by a standard GC-FID) was about 60% smaller than calculated from the total organic OH reactivity (≤ C12). The ozone underprediction for the case of using standard VOC measurements (C2-C8) alone is comparable to the present work. However, the calculated ozone production from the oxidation of VOCs may be overestimated, if an unknown RO$_2$ loss exists as is shown above. In a further study related to observations in the ClearfLo campaign, the comparison of measured and modelled radical concentrations (OH, HO$_2$, RO$_2$#, RO$_2$) points to a significant missing OH source and a missing sink for peroxy radicals at NO concentrations below 1 ppbv (Whalley et al., 2018), which is a similar trend as in the Heshan campaign for OH and RO$_2$. The results of both campaigns indicate significant gaps in the understanding of the radical chemistry and ozone formation in urban air at low NO conditions, which will require further investigations.

## 5 Summary and Conclusions

A field campaign was carried out near the city of Heshan in autumn 2014 studying the radical chemistry under anthropogenically polluted conditions in the Pearl River Delta in southern China. Measurements of radical concentrations (OH, HO$_2$, RO$_2$, RO$_2$#), OH reactivity, and numerous other trace gases were performed. OH reactivity was in the range between 15 s$^{-1}$ and 80 s$^{-1}$, with

median values of 32 s$^{-1}$ in the morning and 22 s$^{-1}$ in the afternoon. Approximately 25% of the reactivity could be explained by measured CO and NOx, another 25% by measured hydrocarbons and formaldehyde, with a remainder of 50% missing reactivity from unmeasured components. OH concentrations reached maximum median values of $4.5 \times 10^6$ cm$^{-3}$ at noon. HO$_2$ and RO$_2$ reached their maximum concentrations later in the afternoon with values of $3 \times 10^8$ cm$^{-3}$ and $2.0 \times 10^8$ cm$^{-3}$, respectively. Measured RO$_2$$^{\#}$ (peroxy radicals mainly from alkenes and aromatics) made up only a small part (15%) of the total RO$_2$, although the fraction of RO$_2$$^{\#}$ producing VOCs made the largest contribution (94%) to the reactivity of measured VOCs. It suggests that at least part of the missing reactivity was due to unmeasured VOCs which produce RO$_2$, but not RO$_2$$^{\#}$.

The diurnal profile of OH was highly correlated with solar radiation and a significant median OH concentration of $0.7 \times 10^6$ cm$^{-3}$ remained after sunset until midnight. In the remaining night, the concentrations dropped below the limit of detection ($0.4 \times 10^6$ cm$^{-3}$, 1$\sigma$). Chemical modulation experiments were performed on several days between noon and midnight in order to test, whether the OH measurements could be biased by artificially produced OH in the low-pressure LIF detection cell. The test experiments were performed at OH reactivities of (14 - 26) s$^{-1}$, NO mixing ratios below 1 ppbv, relative high ozone concentrations (45 - 124 ppbv), and high air temperatures (25 - 30°C). A possible OH interference equivalent to a concentration of $(0.3 \pm 0.3) \times 10^6$ cm$^{-3}$ was found at the limit of detection.

In one of the test experiments, high OH nighttime values (around $1.8 \times 10^6$ cm$^{-3}$) were measured after sunset. These relatively high values are significantly larger than the possible OH interference determined in that test, suggesting that the measured OH was real.

The completeness of the radical measurements at daytime allowed for the first time to perform experimental budget analyses for all radicals (OH, HO$_2$, RO$_2$). There are differences between this method and the analysis often performed in model-based studies. In those studies, turnover rates are calculated from radicals and species that are simulated by a box model. Furthermore, balances between radical production and destruction rates are enforced even if the chemical mechanism is incorrect. In contrast, imbalances in a fully constrained experimental budget analysis, as applied here, indicate directly unknown experimental errors in the input data or an inconsistent chemical mechanism underlying the evaluation.

The balance between radical initiation and termination was studied in the ROx budget. ROx was mainly produced by photolysis of HONO (51%), HCHO (34%) and ozone (15%), and ozonolysis of alkenes (7%). The production with a maximum rate of 3-4 ppbv/h was essentially balanced within 0.5 ppbv/h by the destruction of ROx species with NO or NO$_2$.

In case of RO$_2$, the production rate calculated from measured VOCs is a factor of 4-5 too small to compensate the destruction rate of up to 11 ppbv/h in the afternoon, which is mainly determined by the loss reaction with NO. Only when the missing OH reactivity is explained by unmeasured VOCs can the RO$_2$ loss rate be balanced. The general assumption that missing OH reactivity is equivalent to unmeasured VOCs is thus directly confirmed by RO$_2$ measurements. Although the closure of the RO$_2$ budget is greatly improved, a significant imbalance of (2-5) ppbv/h remains in the afternoon indicating a missing RO$_2$ sink under low NO conditions. As far as RO$_2$$^{\#}$ is concerned, the chemical budget can be quantitatively closed within relatively large experimental error margins, if only measured VOCs are considered for RO$_2$ production. This result implies that the unmeasured VOCs did not produce RO$_2$$^{\#}$ and therefore do not belong to the group of alkenes and aromatics.

The OH destruction is compensated in the morning by the known OH sources from photolysis (HONO, O$_3$), ozonolysis of alkenes and OH recycling (HO$_2$+NO, R9). In the afternoon, however, the destruction rate is significantly higher than the calculated production rate, which indicates a considerable missing OH source of (4-6) ppbv/h. The daily variation of the missing OH source looks similar to that of the missing RO$_2$ sink, but with the opposite sign, so that both compensate each other largely in the ROx budget. Contrary to OH and RO$_2$, the HO$_2$ budget is essentially balanced over the whole day. The difference between

production and destruction rates for OH and $RO_2$ shows an increasing trend when NO falls below 1 ppbv and becomes insignificant above 1 ppbv NO.

The observations indicate the existence of chemical processes that convert $RO_2$ to OH without the involvement of NO. Such processes have been discovered in recent years in the photochemical degradation of isoprene and methacrolein, where OH is regenerated by unimolecular $RO_2$ reactions. However, due to the low abundance of isoprene in the present campaign, these reactions account for only a small fraction ($< 10\%$) of the missing $RO_2$ sink and missing OH source. A generic mechanism has been postulated previously to explain a missing OH source in the PRD under low NO conditions (Hofzumahaus et al., 2009). It involves the successive conversion of $RO_2$ to $HO_2$ and then to OH by an unknown reactant X. A concentration of X equivalent to 0.4 ppbv NO would close the budgets of $RO_2$ and OH in the present study, and leave the budgets of $HO_2$ and ROx unchanged. The X mechanism is, therefore, one possibility to describe all radical budgets at Heshan consistently, although its chemical nature remains unresolved.

The photochemical net ozone production rate was calculated from the reaction rates of $HO_2$ and $RO_2$ with NO, yielding a daily integrated amount of 102 ppbv $O_3$. This amount is due to the oxidation of VOCs (78%), formaldehyde (14%) and CO (8%). About 60% of the ozone production is caused by unmeasured VOCs, which account for half of the measured OH reactivity. An even larger integrated net-ozone production would be calculated from the reaction rate of measured and unmeasured VOCs with OH, if all $RO_2$ radicals would react with NO. However, the unknown $RO_2$ loss reaction removes 22% of the daily $RO_2$ production and thus causes 30 ppbv less ozone production per day than would be expected from the VOC oxidation rate.

In summary, the current work provides new arguments for the existence of a missing OH source, which is most likely due to the conversion of $RO_2$ radicals without the involvement of NO. Our line of arguments depends critically on the assumption that the OH measurement technique is free of artifacts which would erroneously increase the calculated OH loss and $RO_2$ production rates. The experimental tests that were performed in the campaign give no evidence for such an interference, but there remains uncertainty because the tests have not covered the whole period of the campaign. Therefore, further field experiments with continuous control of the LIF measurements by chemical modulation are planned.

**Data availability.**

The data used in this study are available from the corresponding author upon request (a.hofzumahaus@fz-juelich.de).

**Author contributions.**

YZ, AW, and KL organized the field campaign. AH and ZT analyzed the data and wrote the manuscript. All authors contributed to measurements, discussing results, and commenting on the manuscript.

**Acknowledgments**

We thank the science teams of PRIDE-PRD2014. The work was supported by the National Natural Science Foundation of China (**21522701**, **91544225**, **21190052**, **41375124**), the National Key R&D Plan of China (**2017YFC0213000**), the National Science and Technology Support Program of China (**2014BAC21B01**), the Strategic Priority Research Program of the Chinese Academy of Sciences (**XDB05010500**), the Collaborative Innovation Center for Regional Environmental Quality, the BMBF project: ID-CLAR (**01DO17036**) and the EU-project AMIS (Fate and Impact of Atmospheric Pollutants, **PIRSES-GA-2011-295132**).

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

**Table 1 Chemical reactions considered in the radical budget analysis of OH, HO$_2$ and RO$_2$. The radical species are cyclically linked by chain reactions.**

| No. | Reaction | $k$(298 K) [a] |
|---|---|---|
| **Initiation reactions** | | |
| R1 | $HONO + hv\ (< 400nm) \rightarrow OH + NO$ | $j_{HONO}$ [b] |
| R2 | $O_3 + hv\ (< 340nm) \rightarrow O(^1D) + O_2(a^1\Delta_g, X^3\textstyle\sum_g^-)$ | $j_{O1D}$ [b] |
| | $O(^1D) + H_2O \rightarrow OH + OH$ | $2.1\times10^{-10}$ [c] |
| | $O(^1D) + M \rightarrow O(^3P) + M$ | $3.3\times10^{-11}$ [c] |
| R3 | $HCHO + hv\ (< 335nm)\ + 2\ O_2 \rightarrow 2\ HO_2 + CO$ | $j_{HCHO-r}$ [b] |
| R4 | $Alkenes + O_3 \rightarrow OH, HO_2, RO_2 + products$ | [d] |
| **Chain propagation reactions** | | |
| R5 | $OH + RH\ + O_2 \rightarrow RO_2 + H_2O$ | [d] |
| R6 | $HCHO + OH + O_2 \rightarrow CO + H_2O + HO_2$ | $8.4\times10^{-12}$ [c] |
| R7 | $CO + OH + O_2 \rightarrow CO_2 + HO_2$ | $2.3\times10^{-13}$ [c] |
| R8 | $RO_2 + NO \rightarrow RO + NO_2$ | $8.7\times10^{-12}$ [e] |
| | $RO + O_2 \rightarrow R'CHO + HO_2$ | |
| R9 | $HO_2 + NO \rightarrow OH + NO_2$ | $8.5\times10^{-12}$ [c] |
| R10 | $HO_2 + O_3 \rightarrow OH + 2\cdot O_2$ | $2.0\times10^{-15}$ [c] |
| R11 | $NO_2 + hv\ (< 420nm) + O_2 \rightarrow NO + O_3$ | $j_{NO2}$ [b] |
| **Termination reactions** | | |
| R12 | $OH + NO_2 \rightarrow HNO_3$ | $1.1\times10^{-11}$ [f] |
| R13 | $OH + NO \rightarrow HONO$ | $7.5\times10^{-12}$ [f] |
| R14 | $RO_2 + NO \rightarrow RONO_2$ | $4.6\times10^{-13}$ [g] |
| R15 | $RO_2 + RO_2 \rightarrow Products$ | $3.5\times10^{-13}$ [h] |
| R16 | $RO_2 + HO_2 \rightarrow ROOH +\ O_2$ | $2.3\times10^{-11}$ [i] |
| R17[j] | $HO_2 + HO_2 \rightarrow H_2O_2 + O_2$ | $1.7\times10^{-12}$ [c] |
| | $HO_2 + HO_2 + H_2O \rightarrow H_2O_2 + H_2O + O_2$ | $6.4\times10^{-30}$ [k] |

[a] Reaction rate coefficients (cm$^3$ s$^{-1}$) are shown in this table for 298 K and 1 atm. In the radical budget analysis (Fig. 2 - 4), the actual measured ambient temperatures and pressures were used.

[b] Measured (cf.,Table 2) .

[c] MCM3.3.1.

[d] Specific kinetics data for each measured organic compound are taken from MCM3.3.1.

[e] $k(RO_2+NO)= 2.7\times10^{-12}\times exp(360/T)$ (MCM3.3.1). The RO yield is assumed to be 0.95 (see text).

[f] NASA-JPL(Burkholder et al., 2015).

[g] Reaction rate coefficient as for R8. The yield of RONO$_2$ is assumed to be 0.05 (see text).

[h] $k(CH_3O_2+CH_3O_2)=1.03\times10^{-13}\times exp(365/T)$ (MCM3.3.1).

[i] $k(RO_2+HO_2)= f \times 2.91\times10^{-13}\times exp(1300/T)$ (MCM3.3.1). $f$ is a scaling factor which is assumed to be one (see text).

[j] The effective reaction rate $k_{17}$ contains both reactions with and without water.

[k] $k(HO_2+HO_2+H_2O)=3.08\times10^{-34}\times exp(2800/T)+2.59\times10^{-54}\times[M]\times exp(3180/T)$ (RACM2; (Goliff et al., 2013)). This reaction is a termolecular reaction with a unit of cm$^6$ s$^{-1}$.

**Table 2 Median values of measured parameters in the morning and afternoon.**

| | 06:00 -12:00h | 12:00 - 18:00h |
|---|---|---|
| $T$ [°C] | 23.4 | 27.5 |
| $H_2O$ [a] [%] | 2.0 | 2.0 |
| $j_{O1D}$ [$10^{-5}s^{-1}$] | 0.3 | 0.5 |
| $j_{NO2}$ [$10^{-3}s^{-1}$] | 1.6 | 2.1 |
| $j_{HONO}$ [$10^{-4}s^{-1}$] | 2.7 | 3.6 |
| $j_{HCHO-r}$ [$10^{-6}s^{-1}$] | 4.8 | 6.7 |
| OH [$10^6cm^{-3}$] | 1.3 | 2.6 |
| $HO_2$ [$10^8cm^{-3}$] | 0.5 | 2.5 |
| $RO_2$ [$10^8cm^{-3}$] | 0.3 | 1.7 |
| $k_{OH}$ [$s^{-1}$] | 32 | 22 |
| $O_3$ [ppbv] | 16 | 69 |
| NO [ppbv] | 3.7 | 0.4 |
| $NO_2$ [ppbv] | 17 | 9 |
| HONO [ppbv] | 1.1 | 0.4 |
| isoprene [ppbv] | 0.3 | 0.5 |
| HCHO [ppbv] | 5.8 | 6.8 |
| CO [ppmv] | 0.7 | 0.5 |

[a] volume mixing ratio

**Table 3** Unexplained OH signal (mean ± 1σ) and chemical conditions during the OH chemical modulation test experiments. Given numbers are average values of multiple tests that were performed in the specified time period (see as an example, Fig. S3).

| Exp. | Date | Daytime (h) | OH $[10^6\,\mathrm{cm}^{-3}]$ | $k_{OH}$ $[\mathrm{s}^{-1}]$ | $O_3$ [ppbv] | NO [ppbv] | Isoprene [ppbv] | $T$ [°C] | Unexplained OH signal $[10^6\,\mathrm{cm}^{-3}]^a$ |
|---|---|---|---|---|---|---|---|---|---|
| 1 | 19 Oct | 16:40-17:40 | 3.8 | 18.6 | 67 | 0.00±0.11 | N.A. [b] | 27 | 0.7±0.4 |
| 2 | 29 Oct | 14:30-16:00 | 3.7 | 14.1 | 78 | 0.18±0.22 | 0.74 | 29 | 0.3±1.0 |
| 3 | 30 Oct | 20:20-21:00 | 1.8 | 25.7 | 32 | 0.21±0.25 | 0.04 | 25 | 0.4±0.5 |
| 4 | 31 Oct | 12:50-13:50 | 9.4 | 20.1 | 103 | 0.32±0.38 | 0.60 | 30 | 0.1±0.8 |
| 5 | 1 Nov | 15:10-16:00 | 6.9 | 22.2 | 124 | 0.10±0.05 | 1.21 | 30 | 0.4±0.7 |
| 6 | 22 Nov | 17:00-23:00 | 0.2 | 24.4 | 45 | 0.14±0.24 | 0.05 | 25 | -0.3±0.5 |

[a] Expressed as equivalent OH concentration; [b] Not available.

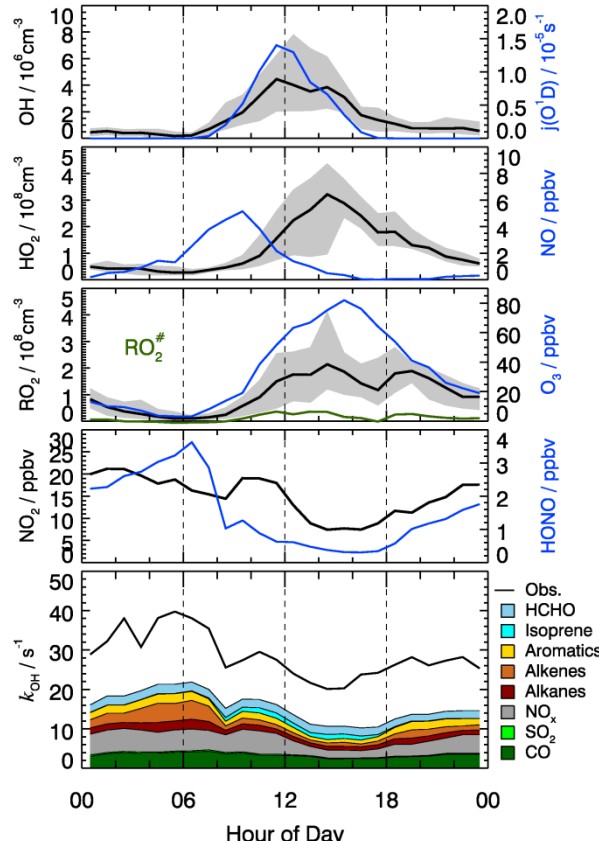

**Figure 1 Median diurnal profiles of measured OH, HO$_2$, RO$_2$, RO$_2^{\#}$, $k_{OH}$, $j(O^1D)$, NO, NO$_2$, O$_3$, and HONO. For OH, HO$_2$ and RO$_2$, the grey band around the median (black lines) denotes the 25 % and 75 % percentiles of the distributions. In the lowest panel, coloured areas show the speciated reactivity contributions from measured OH reactants.**

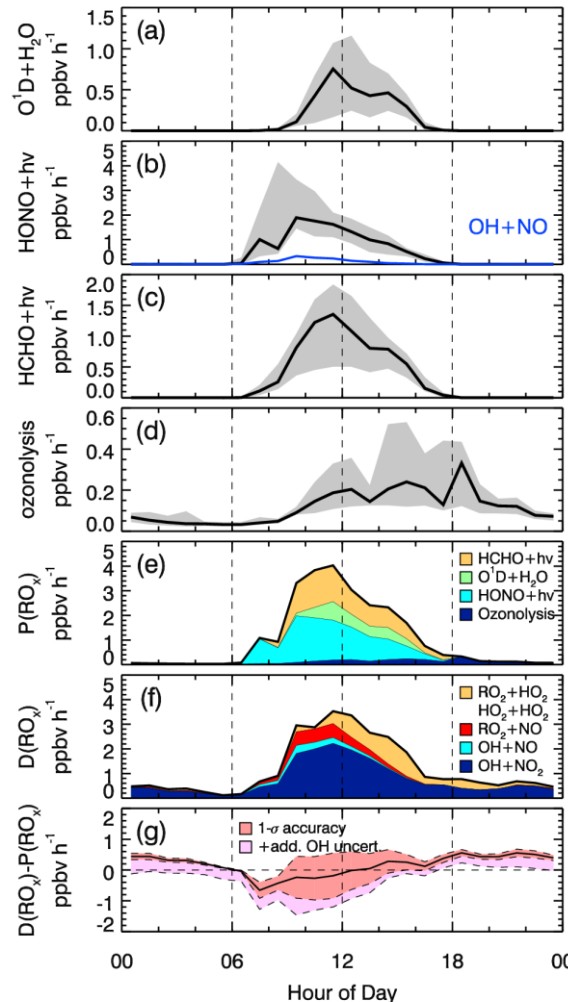

**Figure 2 Median production and destruction rates of RO$_x$.** Panels (a - d) show production rates (black lines: median values) from the photolysis of ozone (a), HONO (b), HCHO (c), and ozonolysis of alkenes (d). Grey bands around the median denote 25 % and 75 % percentiles. The blue line in panel (b) represents the back reaction rate of OH+NO yielding HONO. Panels (e) and (f) show cumulative plots of the production and destruction rates, respectively. Panel (g): the solid black line is the difference between the total production and destruction rate. The red shaded band indicates the 1σ uncertainty due to experimental errors of the measured quantities (Table S1) and the reaction rate coefficients. The pink shaded area represents the maximum possible bias from a potential OH interference.

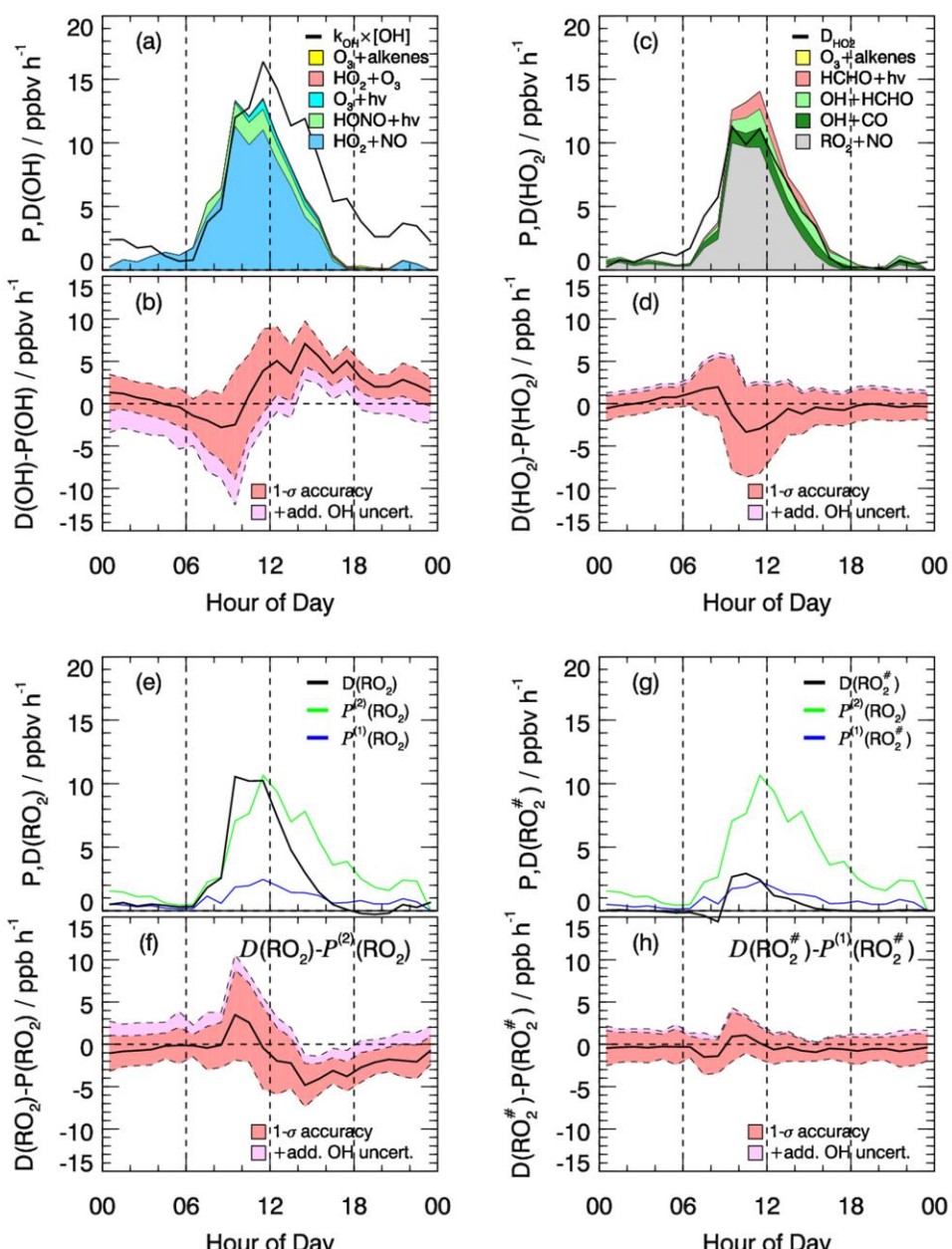

**Figure 3 Experimental budgets for OH (a, b), HO₂ (c, d), RO₂ (e, f) and RO₂# (g, h). In the respective upper panels (a, c, e, g), solid black lines denote the median total destruction rates. The colored areas in (a) and (c) represent cumulative plots of the production rates from different reactions. The blue solid lines in (e) and (g) denote the production rates $P^{(1)}_{RO2}$ and $P^{(1)}_{RO2\#}$, respectively, calculated from measured VOCs (Equation E7). The green lines represent $P^{(2)}_{RO2}$ calculated from $k_{OH}(VOC(2))$ (Equation E8). In all four budgets (OH, HO₂, RO₂, RO₂#) the radical production from ozonolysis is hardly noticeably small. The respective lower panels (b, d, f, h) show the difference between the total destruction and production rates. Red shaded bands indicate the 1σ uncertainty due to experimental errors of the measured quantities (Table S1) and the reaction rate coefficients. The pink shaded areas represent the maximum possible bias from a potential OH interference.**

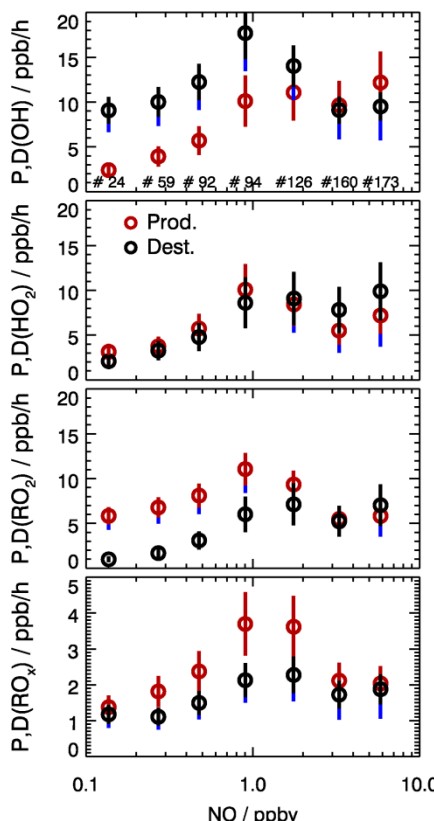

**Figure 4 Experimental production and destruction rates for OH, HO$_2$, and RO$_2$ as a function of NO. The circles represent median values for NO intervals of △ln(NO)/ppbv=0.57. Data are restricted to daytime conditions (j(O$^1$D)>0.1×10$^{-5}$ s$^{-1}$). The number (#) of data points included in each NO bin is given in the upper panel. Vertical error bars (red and black) denote the 1σ uncertainties due to experimental errors of the measured quantities (Table S1) and the reaction rate coefficients. Vertical blue bars denote the maximum possible bias from a potential OH interference.**

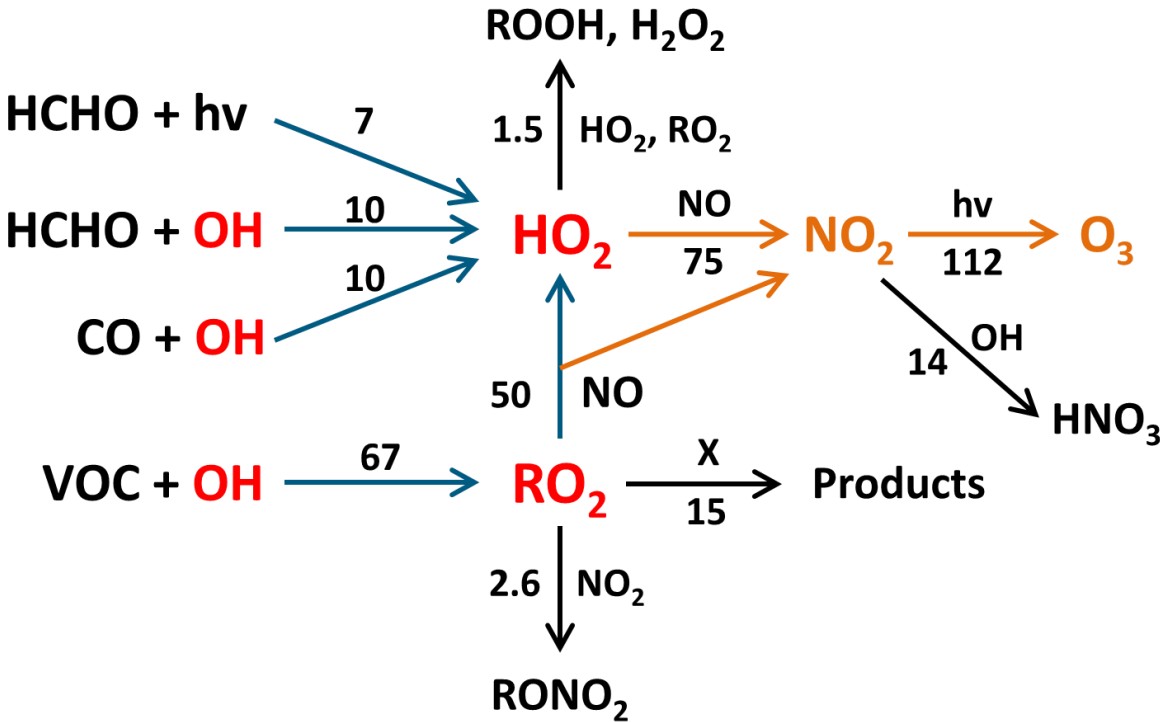

**Figure 5 Main reaction pathways leading to photochemical net ozone formation during the Heshan campaign. Blue arrows show the reaction paths leading to $RO_2$ and $HO_2$, yellow arrows indicate the oxidation reactions of NO to $NO_2$ by peroxy radicals with subsequent $NO_2$ photolysis yielding ozone. Black arrows represent reactions that remove peroxy radicals or $NO_2$ and thus reduce ozone production. Numbers represent median daily-integrated reaction rates (ppbv) calculated by equation E12. The value for VOC+OH corresponds to kOH(VOC(2))×[OH] (see equation E8). The reaction with X indicates an unknown process that removes $RO_2$ without producing $NO_2$.**