# Peer review of "Experimental budgets of OH, HO2 and RO2 radicals and implications for ozone formation in the Pearl River Delta in China 2014"

_Atmospheric Chemistry and Physics, 2018_

## Referee Comment (RC1) · Anonymous Referee #2 · 9 Oct 2018

This manuscript presents an extensive dataset of radical measurements in the Pearl River Delta region (China). The authors use these data to examine the budget of OH, HO2 and total RO2. There are not many co-located measurements of OH, HO2 and RO2, so this dataset and its analysis provide a rather unique and interesting look into the chemistry of the polluted atmosphere. The presentation of the data is well laid out and the discussion is clear. I have a few questions and comments for the authors, but, other than that, I recommend publication in ACP.

page 4, lines 19-20. "The interference is most effective when the amount of added NO is sufficiently high to convert most of the atmospheric HO2 to OH in the LIF cell" This

sentence is followed by the statement that the concentration of NO was reduced by a factor of 10. Does this mean that the conversion of HO2 to OH in the HO2 cell is not complete? Could this lead to underestimation of HO2?

page 5, lines 32-34. "The main reactants are NO and the peroxy radicals themselves, all of which were measured allowing the total loss rates from the individual reactions to be calculated." I am not sure this statement is correct. The ROxLIF technique certainly provides new information, but it still measures the sum of peroxy radicals, so I don't think the authors can claim that all of the peroxy radicals were measured.

The assumption that all RO2 have similar rate coefficients (with HO2, other RO2 and/or NO) is a very common one, but it is still a rather big assumption. The MCM itself uses two different generic rate coefficients for RO2+RO2, RO2+NO and RO2+HO2 reactions, depending on the type of peroxy radical. This issue is also mentioned on pages 6 and 7, and may be relevant for the discussion of the RO2 budget in Section 3.7. Moreover, if part of the argument is that the RO2 budget is closed within the instrumental uncertainty, but still slightly negative (page 12) than this could be a factor to consider. The authors correctly discuss on page 13 how the assumptions on the nitrate yields, which are a similar issue, affect the conclusions of the paper. But I don't see a similar discussion for the rate coefficients.

page 7. The two methods to calculate RO2 production from OH+VOC reactions take into consideration the possibile effect of unmeasured VOC, which is correct. However, a similar approach was not taken with regard to unmeasured alkene that react with ozone. Such missing VOC may be an issue for the calculation of OH sources in E4 and the discussion of the HO2 budget in Section 3.6. The potential problem is acknowledged on page 6, but there is no discussion of how it may affect the conclusions of the paper.

page 12, lines 1-2. It is not clear if the authors are discarding the hypothesis of Yang et al (2017) that the missing reactivity is at least partly due to OVOC and, if so, why.

[Figure]

Figure 3, panels f and g. In one panel the difference between destruction and pro-
duction is compared to that derived from VOC(1) and in the other is compared to that
derived from VOC(2). I see what the authors are trying to do, but it is a bit misleading.
Maybe both differences could be shown by adding a third panel for both RO2 and RO2#
or maybe different colors could be used.

page 11, line 25. Typo: "by developed".

---

## Referee Comment (RC2) · Anonymous Referee #1 · 15 Oct 2018

This paper presents an analysis of OH, HO2, and RO2 radical budgets based on direct measurements of these radicals as well as measurements of their sources and sinks during the PRIDE-PRD2014 (Program of Regional Integrated Experiments of Air Quality over the Pearl River Delta) in 2014. Because this study involved direct measurements of the concentration of each radical family, the authors are able to provide an analysis of the radical budgets based on measured concentrations in contrast to previous studies that used the results of chemical models to estimate concentrations of unmeasured radical concentrations. The results illustrate that while the radical budgets for HO2 and total ROx are balanced to within the uncertainty of the measurements, the budget for RO2 radicals can only be balanced if the observed missing OH reactivity

is due to reaction of unmeasured VOCs leading to the production of RO2 radicals. In addition, the analysis suggests a missing RO2 sink and a missing OH source in the afternoon which cancel each other in the total ROx budget. The budgets could be closed by including an additional chemical mechanism that converts RO2 radicals to OH that does not involve reaction with NO. While an interference with their measurements of OH could explain these discrepancies, the authors provide some evidence that their measurements were free of unknown interferences. However, they acknowledge that additional measurements with more continuous testing for interferences are needed to confirm these results. The authors use the measurements to calculate integrated net rates of ozone production, and find that a missing RO2 sink would reduce ozone production by approximately 30% compared to that expected from the VOC oxidation rate.

The paper is well written and provides a new perspective on our understanding of radical chemistry. It is suitable for publication in ACP after the authors have addressed the following minor comments.

1) The authors provide evidence that the missing OH reactivity is due to missing VOCs that react to produce RO2 radicals. While this provides strong evidence that the missing OH reactivity is not due to radical termination reactions, it is not clear from the discussion on pages 11 and 12 that this rules out that the missing reactivity is due to unmeasured OVOCs as discussed by Yang et al. (2017). This should be clarified.

2) One of the main conclusions of the paper is that the budget analysis suggests a missing RO2 radical sink and an OH radical source in the afternoon. While the authors suggest that RO2 isomerization reactions from isoprene and methacrolein may not be important given the low concentrations of isoprene measured, there may be other autoxidation processes that could be important in the afternoon when NO concentrations are low (see Praske et al., PNAS, 115, 64-69, 2018). This should be discussed in more detail.

**ACPD**
3) The authors provide some evidence that their OH measurements are free from interferences through some chemical modulation tests. While the majority of these measurements appear to be below the detection limit for the instrument, it is not clear from Table 3 whether the data presented represent an average of multiple tests during the time period indicated, or a single modulation experiment. It would useful clarify the number and duration of the modulation experiments, perhaps by showing some of the raw data from the experiments in the supplement.

---

## Referee Comment (RC3) · Anonymous Referee #3 · 17 Oct 2018

The article "Experimental budgets of OH, HO2 and RO2 radicals and implications for ozone formation in the Pearl River Delta in China" by Tan et al. analyses the radical budgets for a polluted region utilising comprehensive observations of radical species, total OH reactivity and radical precursors. The total budgets of ROx and OH, HO2, RO2 are compared pointing to a conversion between ROx to HO2 and OH by an unknown component X as formerly hypothised by Hofzumahaus et al. 2009, in the Pearl River Delta and unmeasured VOC species. The implications for the O3 production potential are discussed to be less effective as assumed by total OH reactivity measurements. The article is well written with an an unique dataset, calculating the budgets with a measurement approach only. I recommend publication after following remarks have

been addressed:

page1, line 17: Typo: "radical interconversion" instead of "radical interconveresio"

page 1, line 19: "In case of RO2, the budget can only be closed when the missing OH reactivity is attributed to unmeasured VOCs. Thus, the existence of unmeasured VOCs is directly confirmed by RO2 measurements." This is a likely but not exclusive explanation. I recommend to rephrase the sentences: "In case of RO2, the budget could be closed by attributing the missing OH reactivity to unmeasured VOCs. Thus, unmeasured VOCs are directly linked to the RO2 measurements."

page 1, line 25: "These observations suggest the existence of a chemical mechanism that converts RO2 to OH without the involvement of NO. Please quantify the average contribution of this channel to the total turnover rate.

page 2, line 17: Please explain the abbreviation of "PRIDE-PRD" already here.

page 2, line 36: " ... tendency to underpredict OH under low-NOx conditions" . Please define the term "low-NOx conditions".

Page 4, line 4: Please explain in the supplement table caption the term "RO2#".

Page 4, line 12: Please quantify how much OH is internally removed.

Page 4, line 13: How did you account for possible impurities in N2?

Page 4, line 21: Please quantify the upper limit of the HO2 interference .

Page 4, line 33: How did you quantify the HO2 background signal?

Page 5, line 8: It is unclear which instruments have been used for CO and CO2 in this study.

Page 6, line 36: Please quantify the impact of the assumption that the nitrate yield is 5%. Please add a reference for the nitrate yield.

Page 7, line 17: Despite the fast NO3 photolysis Liebmann et al. , 2018, found during

daytime a fractional loss of NO3 of 25% by reaction with BVOC. What is the daytime NO3 production rate in this study and what would be an upper limit for its contribution ?

Page 9, line 38, What is the upper limit of ROx production by NO3 during night time, i.e. what is the NO3 production rate ?

Page 10, line 24 Is it the only exclusive explanation or a possible explanation that fits the result ?

Page 11, line 6: "The completeness of the radical measurements allows a budget analysis for all radicals (OH, HO2, RO2) based on experimental data only, ...". Please add: "under the assumption that for the production and loss rates all relevant species were measured."

Page 11, line 9: How do you define daytime?

Page 11, line 25: typo : "... comparative method developed by ..."

Page 11, line 30: It is not obvious that under NOx regimes, controlling radical propagation and termination schemes, the resulting intermediates or even the emitted VOC found in Yang et al are comparable with the ones in this study.

Page 12, line 4: Please specify uncertainties.

Page 13, line 35 Please quantify "negligible" including upper limit for dry deposition

Page 13, line 36&37; Page 14 line 7 : Specify uncertainties

Page 14, line 4: Please include the statement that the loss term HO2*NO generating NO2 can be replaced by the production term of HO2 under the assumption that other HO2 losses, like HO2+RO2, HO2+HO2 are negligible.
* * *

---

## Author Response (AR1)

**Response to reviewers and changes made to the manuscript**

We would like to thank all reviewers for their questions and comments, which have helped us to improve the manuscript. Below you find comments to the editor and our detailed answers to the comments of the reviewers. Referee comments are given in italics, our responses are in normal font. Changes made to the manuscript are marked in blue.

*Comments to the editor*

We corrected a few minor errors which we discovered during the revision of the manuscript and we added clarifications in the text. The corrections and changes do not affect the results and conclusions of the paper.

(1)

On page 14 of the original manuscript (line 18-19), the values of 15% and 30% must read 15 ppbv and 30 ppbv, respectively. The corrected sentence reads:
In the present case, such an $RO_2$ sink would remove 15 ppbv (22%) of the daily produced $RO_2$ resulting in an integrated ozone production that is 30 ppbv lower than expected from the rate of VOC oxidation. Likewise, we corrected the corresponding sentences in the abstract and conclusion section.

(2)

For better understanding of the daily ozone production at Heshan, we added an illustration. The new Figure 5 shows the major reaction pathways and the corresponding reaction rates leading to ozone formation.

[Figure]

**Figure 5 Main reaction pathways leading to photochemical net ozone formation during the Heshan**

**campaign. Blue arrows show the reaction paths leading to RO₂ and HO₂, yellow arrows indicate the oxidation reactions of NO to NO₂ by peroxy radicals with subsequent NO₂ photolysis yielding ozone. Black arrows represent reactions that remove peroxy radicals or NO₂ and thus reduce ozone production. Numbers represent median daily-integrated reaction rates (ppbv) calculated by equation E12. The value for VOC+OH corresponds to kOH(VOC(2))×[OH] (see equation E8). The reaction with X indicates an unknown process that removes RO₂ without producing NO₂.**

(3)

In Figure 2g (original manuscript), $D$(ROx)-$P$(ROx) was incorrectly calculated as the difference of the medians of $D$(ROx) and $P$(ROx). In the revised version, we have recalculated $D$(ROx)-$P$(ROx) as the median of the difference of $D$(ROx) and $P$(ROx), which is more correct. Due to the recalculation, the numerical balance between production and destruction rates improved slightly. However, given the experimental uncertainties, the improvement plays a minor role. The conclusions drawn from the figure remain unchanged.

[Figure]

On page 9 (line 32-34), the description of the figure was revised and a statement about the possible role of NO₃ chemistry for the ROx budget at night was added (see our response to question 12 by Referee #3). During daytime, there is an imbalance between production ($P_{ROX}$) and destruction ($D_{ROX}$) rates, which increases from -0.5 ppbv (8:00) to +0.5 ppbv (18:00). The deviations are small ($< 15\%$) compared to the total ROx turnover rate at noontime and can be explained by the experimental errors of the data between 8:30 and 17:30. After sunset until 2 hours after midnight, $D_{ROx}$ remains about 0.5 ppbv/h larger than $P_{ROx}$. The deviation may be caused, at least partly, by RO₂ formation from reactions of VOCs with NO₃ radicals, for which an upper limit of 0.7 ppbv/h can be estimated (see Section 4.2). The deviation can also largely be explained by experimental errors, when the upper limit of a possible OH interference is included in the total experimental uncertainty (Fig. 2g).

**Anonymous Referee #1**

*Comments*

(1)

*The authors provide evidence that the missing OH reactivity is due to missing VOCs that react to produce RO$_2$ radicals. While this provides strong evidence that the missing OH reactivity is not due to radical termination reactions, it is not clear from the discussion on pages 11 and 12 that this rules out that the missing reactivity is due to unmeasured OVOCs as discussed by Yang et al. (2017). This should be clarified.*

We do not rule out that unmeasured OVOCs may be responsible for the missing OH reactivity and produce RO$_2$ by reaction with OH. Many oxygenated VOCs (ketones, C2- and higher aldehydes, C3- and higher alcohols) produce RO$_2$ when they react with OH. Examples are acetaldehyde, acetone, MACR, MVK, methyl glyoxal etc. Other OVOCs like formaldehyde, glyoxal etc. do not produce RO$_2$.

For clarification, we have modified the text (bottom of page 11 and top of page12) as follows.

Many oxygenated VOCs produce RO$_2$ when they react with OH. Examples are acetaldehyde and higher aldehydes, acetone and higher ketones, MACR, MVK, and methyl glyoxal, all of which were not measured in the present study. In the following discussion we assume based on the considerations given above that the missing reactivity in the present study is entirely due to VOCs (including also OVOCs) which can produce RO$_2$ by reaction with OH.

(2)

*One of the main conclusions of the paper is that the budget analysis suggests a missing RO$_2$ radical sink and an OH radical source in the afternoon. While the authors suggest that RO$_2$ isomerization reactions from isoprene and methacrolein may not be important given the low concentrations of isoprene measured, there may be other autoxidation processes that could be important in the afternoon when NO concentrations are low (see Praske et al., PNAS, 115, 64-69, 2018). This should be discussed in more detail.*

We agree that this topical aspect deserves a deeper discussion. In fact, there is a fast growing body of literature reporting experimental and theoretical results about autoxidation reactions which involve H-migration in RO$_2$ species at atmospheric conditions that can possibly regenerate OH radicals at low NO conditions. We extended the discussion in Section 4.2. The revised section reads as follows.

[revised manuscript text omitted]

The caption of Figure S3 (now S4) was revised as follows.

Figure S4: Same as Fig. 3, but with additional $RO_2$ conversion to OH assuming a first-order rate coefficient of 0.08 s$^{-1}$. This scenario can also be seen as an application of the X mechanism which recycles OH by the hypothetical sequence $RO_2 + X \rightarrow HO_2$, $HO_2 + X \rightarrow HO_2$ with X equivalent to 0.4 ppbv NO.

(3)

*The authors provide some evidence that their OH measurements are free from interferences through some chemical modulation tests. While the majority of these measurements appear to be below the detection limit for the instrument, it is not clear from Table 3 whether the data presented represent an average of multiple tests during the time period indicated, or a single modulation experiment. It would useful clarify the number and duration of the modulation experiments, perhaps by showing some of the raw data from the experiments in the supplement.*

The results presented in Table 3 are an average of multiple tests during the specified time period. We have added the following explanation in the caption of Table 3 and included as an example a new Figure S3 in the Supplement.

Given numbers are average values of multiple tests that were performed in the specified time period (as an example, see Fig. S3).

[Figure]

**Figure S3 Results from the chemical modulation tests performed on 31 October 2014 between 12:50 and 13:50. The measured OH signal without scavenger ($S_{N2}$) can be explained within experimental errors by the sum of the signal from ambient OH ($S_{OH}$) and the known interference from $O_3$ ($S_{O3}$). Error bars denote $1\sigma$ statistical errors. $S_{OH}$ is calculated by the expression ($S_{N2}$ - $S_{propane}$)/$\varepsilon$, where $S_{propane}$ is the signal with scavenger (propane) and $\varepsilon$ is the efficiency of scavenging (for details, see Tan et al., 2017). A fluorescence signal of 60 cts/s is equivalent to an OH concentration of $1 \times 10^7$ cm$^{-3}$.**

**References:**

[revised manuscript text omitted]
" This sentence is followed by the statement that the concentration of NO was reduced by a factor of 10. Does this mean that the conversion of $HO_2$ to OH in the $HO_2$ cell is not complete? Could this lead to underestimation of $HO_2$?*

The reduction of NO does not lead to an underestimation of the measured $HO_2$ concentration. The sensitivity of the $HO_2$ measurement channel depends on many instrumental parameters, one of which is the $HO_2$ conversion efficiency. The overall sensitivity is experimentally determined by the calibration, which determines the fluorescence signal for a given $HO_2$ concentration provided by the calibration source.

(2)

*page 5, lines 32-34. "The main reactants are NO and the peroxy radicals themselves, all of which were measured allowing the total loss rates from the individual reactions to be calculated." I am not sure this statement is correct. The ROxLIF technique certainly provides new information, but it still measures the sum of peroxy radicals, so I don't think the authors can claim that all of the peroxy radicals were measured.*

*The assumption that all $RO_2$ have similar rate coefficients (with $HO_2$, other $RO_2$ and/or NO) is a very common one, but it is still a rather big assumption. The MCM itself uses two different generic rate coefficients for $RO_2+RO_2$, $RO_2+NO$ and $RO_2+HO_2$ reactions, depending on the type of peroxy radical. This issue is also mentioned on pages 6 and 7, and may be relevant for the discussion of the $RO_2$ budget in Section 3.7. Moreover, if part of the argument is that the $RO_2$ budget is closed within the instrumental uncertainty, but still slightly negative (page 12) than this could be a factor to consider. The authors correctly discuss on page 13 how the assumptions on the nitrate yields, which are a similar issue, affect the conclusions of the paper. But I don't see a similar discussion for the rate coefficients.*

We agree with the referee in both points. The ROxLIF technique is measuring the sum of $RO_2$ radicals and the technique is not equally sensitive to all $RO_2$ radical species. In the Supplementary Text, we added an explanation how this measurement bias influences our budget analysis (see below). The second point is also true. $RO_2$ radicals have different rate constants for their reaction with NO, $HO_2$, and $RO_2$. The error

that comes from the use of effective rate coefficients for the lumped $RO_2$ in Table 1 will be addressed in the revised text and explained in the Supplementary Text.

In the main paper, Section 4.2 was renamed to "4.2 Radical budgets, their relationships and uncertainties" and the following text was added on page 13.

Another uncertainty is caused by the measurement and incomplete representation of the $RO_2$ chemistry. Due to the measurement principle of the ROxLIF instrument, only those $RO_2$ species are measured which are converted in the instrument to $HO_2$ by reaction with NO. This measurement is suitable to quantify the $HO_2$ production rate (Equation E5). Among the $RO_2$ radicals which are not completely captured by ROxLIF are species which produce a new $RO_2$ radical when they react with NO. As these reactions are neutral with respect to the total amount of $RO_2$ , the $RO_2$ budget ($D$-$P$) is not sensitive to the bias of the $RO_2$ measurement caused by these species (see Supplementary Text).
Other uncertainties in the $RO_2$ budget are caused by the rate constants for the reactions of $RO_2$ with NO (Reaction R8, R14), $RO_2$ (Reaction R15), and $HO_2$ (Reaction R16) that are given in Table 1 as effective values for the lumped $RO_2$ radicals. In this work, the uncertainties of the rate coefficients for Reaction R15 and R16 play only a minor role, because the daytime loss of the peroxy radicals was largely dominated by the reaction with NO (see Supplementary Text). The relevant range for the reaction rate constants of different $RO_2$ species with NO (Reaction R8, R14) is between $8 \times 10^{-12}$ $cm^3 s^{-1}$ and $1.1 \times 10^{-11}$ $cm^3 s^{-1}$ (see Supplementary Text). As a sensitivity test, Figs. S5 and S6 show the budgets of ROx, $RO_2$ and $HO_2$ for a rate constant of $1 \times 10^{-11}$ $cm^3 s^{-1}$. The results are essentially the same as in Figs. 2 and 3 where a rate constant of $9 \times 10^{-12}$ $cm^3 s^{-1}$ was applied for Reaction R8 + R14. Thus, an increased rate constant cannot explain the missing $RO_2$ sink in the $RO_2$ budget.

In the Supplementary Text, the following text was included.

*Uncertainties related to the measurement and chemistry of $RO_2$*
Uncertainties in the radical budgets may be caused by the measurement and incomplete representation of the $RO_2$ chemistry. Due to the measurement principle of the applied ROxLIF technique, only those $RO_2$ species can be measured which are converted to $HO_2$ by reaction with NO for conditions of the ROxLIF system. This measurement is exactly what is needed to quantify the $HO_2$ production rate (Equation E5) in the atmospheric $HO_2$ budget. However, using the measured $RO_2$ data for the calculation of the $RO_2$ loss rate (Equation E9) may cause a systematic bias. $RO_2$ radical species exist which react with NO and produce a new $RO_2$ radical rather than $HO_2$. An example is the reaction $(CH_3)_3C(O_2)+NO$ leading to $CH_3O_2$+acetone+$NO_2$ as products. The result is a low-biased measurement of atmospheric $RO_2$ radicals. Its use in Equation E9 leads to an underestimation of $D_{RO2}$ since the $RO_2$ loss leading to new $RO_2$ species is not included due to the measurement bias. On the other side, the production $P_{RO2}$ in equation E8 is underestimated by the same amount, because the production term for $RO_2$ species which are produced by $RO_2$+NO is missing. As a result, the balance term $D_{RO2}$-$P_{RO2}$ in Fig. 2 remains correct as the production and destruction terms are smaller by the same unknown amount. Another group of $RO_2$ radicals which are not well captured by ROxLIF are nitrate peroxy radicals, which are formed by the reaction of $NO_3$ radicals with alkenes. Some nitrate peroxy radical species (e.g., from propene and butenes) react with NO and produce carbonyl compounds and $NO_2$ as products. The latter reaction constitutes an ROx sink. In the present work, $NO_3$ reactions with VOCs play a minor role (Section 4.2).
Other uncertainties in the $RO_2$ budget are caused by the rate constants that are given in Table 1 as effective values for the lumped $RO_2$ radicals. It is well known that the rate coefficients for the reactions of

RO$_2$ with NO, HO$_2$, and RO$_2$ depend on the chemical structure of the RO$_2$ species. According to Jenkin et al. (2019), experimentally known rate constants for RO$_2$+NO can be broadly categorized into three classes: [1] CH$_3$O$_2$ (C1), [2] other hydrocarbon ($\geq$ C2) and oxygenated peroxy radicals, and [3] acyl peroxy radicals. At room temperature, recommended rate constants for these categories are $7.7\times10^{-12}$ cm$^3$s$^{-1}$, $9.0\times10^{-12}$ cm$^3$s$^{-1}$, and $2.0\times10^{-11}$ cm$^3$s$^{-1}$, respectively (Jenkin et al., 2019). The MCM value used in Table 1 for Reaction R8 + R14 ($9.0\times10^{-12}$ cm$^3$s$^{-1}$) fits to the second class. The high rate constants for acyl peroxy radicals have no relevance for the budget analysis, because their reaction with NO produces another RO$_2$ radical. Thus, their reaction does not contribute to the HO$_2$ production and is neutral in the RO$_2$ budget as explained above. Published rate constants of the second category range between $8\times10^{-12}$ cm$^3$s$^{-1}$ and $1.1\times10^{-11}$ cm$^3$ s$^{-1}$ (Jenkin et al., 2019). Here, the lower limit is almost equal to the rate coefficient of CH$_3$O$_2$ (first class). As a sensitivity test, Figs. S5 and S6 show the budgets of ROx, RO$_2$ and HO$_2$ for a rate constant of $1\times10^{-11}$ cm$^3$s$^{-1}$ (R8 + R14). The results are essentially the same as in Figs. 2 and 3 where a rate constant of $9\times10^{-12}$ cm$^3$s$^{-1}$ is applied. As the RO$_2$ budget indicates a missing RO$_2$ sink, a larger rate constant could help resolve the discrepancy. However, the 10% increase of the rate constant for Reaction R8 + R14 in Figs. S5 and S6 is far too small to explain the observed imbalance.

The reaction of RO$_2$ radicals with NO can form HO$_2$ (Reaction R8) resulting in radical chain propagation, or produce organic nitrates (Reaction R14) resulting in chain termination. As the branching ratio can be different for each RO$_2$ species and as most of the organic reactivity was caused by unmeasured VOCs, the branching ratios of most RO$_2$ species are not known. Typical yields for organic nitrates lie in the range between 1% and 35% (Atkinson et al., 1982; Lightfoot et al., 1992). For the budget analysis (Figs. 2 - 4), an organic nitrate yield of 5% is assumed. Figs. S7 and S8 show cases where higher yields (10%, 20%) are assumed. Higher organic nitrate yields compensate the slightly negative bias of *D-P* in the ROx budget (Fig. S7). An average yield of 10% would lead to similar difference between production and destruction rate of ROx during daytime, whereas a yield of 20% would result in a slightly positive bias of up to +1 ppbv/h in *D-P*. For the HO$_2$ production rate, these changes have little impact. Thus, in all cases (80%, 90%, 95% yield of HO$_2$), the HO$_2$ budget is balanced within the experimental uncertainties.

Published rate constants for the reaction RO$_2$+HO$_2$ (Reaction R16) lie in the range between $0.5\times10^{-11}$ cm$^3$s$^{-1}$ and $2.2\times10^{-11}$ cm$^3$s$^{-1}$ at 298K (Jenkin et al., 2019). In the MCM, a general value of $2.3\times10^{-11}$ cm$^3$s$^{-1}$ (298K) is assumed and scaled by an RO$_2$ specific factor which is typically 0.5 - 0.7. In the budget analysis we have used the upper limit with a scaling factor of one. Thus, the possible bias of the calculated RO$_2$+HO$_2$ rate is in the order of a factor of 2. Under the polluted conditions of the campaign, the loss of RO$_2$ and HO$_2$ is largely dominated by NO. The reaction RO$_2$+HO$_2$ contributes only a few percent to the ROx loss during daytime and no more than 10% at sunset, when NO is small. Thus, the bias in the calculated ROx loss rate remains well below 5% at daytime. Similar considerations apply to the loss of RO$_2$ and HO$_2$, which is also dominated by NO during the day.

Rate coefficients for self and cross reactions of RO$_2$ are diverse and difficult to parameterize (Jenkin et al., 2019). The rate constants for the most abundant species are generally an order of magnitude smaller than for the reaction R16 (RO$_2$+HO$_2$). Self reactions of oxygenated RO$_2$ and cross reactions of some RO$_2$ can be as fast as reaction R16 (Jenkin et al., 2019). Overall, RO$_2$+RO$_2$ reactions play a smaller role than RO$_2$+HO$_2$ reactions in the Heshan campaign. The uncertainty of the RO$_2$ radical budget due to the lumped rate coefficient for R15 is therefore negligible.

[Figure]

**Figure S5** Same as Figure 2, but assuming a rate constant of $1\times10^{-11}cm^{-3}s^{-1}$ for the reaction of $RO_2$ with NO (R8, R14) .

[Figure]

**Figure S6** Same as Figure 3, but assuming a rate constant of $1\times10^{-11}cm^{-3}s^{-1}$ for the reaction of $RO_2$ with NO (R8, R14) .

(3)

*page 7. The two methods to calculate RO2 production from OH+VOC reactions take into consideration the possible effect of unmeasured VOC, which is correct. However, a similar approach was not taken with regard to unmeasured alkenes that react with ozone. Such missing VOC may be an issue for the calculation of OH sources in E4 and the discussion of the HO2 budget in Section 3.6. The potential problem is acknowledged on page 6, but there is no discussion of how it may affect the conclusions of the paper.*

The budget analysis for $RO_2$# suggests that there were not many more alkenes present besides the measured ones. We mentioned this point in the original version of the Supplementary Text, but obviously our explanation was not clear. We extended the explanation in the Supplementary Text as follows. Information about the abundance of alkenes in this campaign can be obtained from the $RO_2$# budget analysis. $RO_2$# is produced by OH reaction with alkenes, aromatics and large alkanes. The budget analysis (Fig. 3) shows that the calculated production rate $P^{(1)}_{RO2\#}$ from these compounds is balanced by the calculated $RO_2$# loss rate. If an essential fraction of the unmeasured VOCs consisted of alkenes, it would increase the $RO_2$# production rate correspondingly. Within experimental uncertainty, a doubling of the alkene contribution in the $RO_2$# production would be acceptable without disturbing the balance in the $RO_2$# budget. Doubling of the alkenes would explain 15% of the missing OH reactivity. In this case, the radical production from ozonolysis, which is less than 0.1 ppbv/h for OH and 0.05 ppbv/h for $HO_2$ at daytime taking measured species into account, would increase by about a factor of 2. This increase would have a negligible impact on the radical budgets of OH and $HO_2$.

(4)

*page 12, lines 1-2. It is not clear if the authors are discarding the hypothesis of Yang et al (2017) that the missing reactivity is at least partly due to OVOC and, if so, why.*

The same question was raised by referee #1. See our response there.

(5)

*Figure 3, panels f and g. In one panel the difference between destruction and production is compared to that derived from VOC(1) and in the other is compared to that derived from VOC(2). I see what the authors are trying to do, but it is a bit misleading. Maybe both differences could be shown by adding a third panel for both RO2 and RO2# or maybe different colors could be used.*

For clarification, we modified the legends in panels (e) and (g), and we added consistent labels in panels (f) and (g), which make direct reference to Equations E7 and E8.

[Figure]

Figure 3 Experimental budgets for OH (a, b), HO$_2$ (c, d), RO$_2$ (e, f) and RO$_2^{\#}$ (g, h). In the respective upper panels (a, c, e, g), solid black lines denote the median total destruction rates. The colored areas in (a) and (c) represent cumulative plots of the production rates from different reactions. The blue solid lines in (e) and (g) denote the production rates $P^{(1)}_{RO2}$ and $P^{(1)}_{RO2\#}$, respectively, calculated from measured VOCs (Equation E7). The green lines represent $P^{(2)}_{RO2}$ calculated from $k_{OH}(VOC(2))$ (Equation E8). In all four budgets (OH, HO$_2$, RO$_2$, RO$_2^{\#}$) the radical production from ozonolysis is hardly noticeably small. The respective lower panels (b, d, f, h) show the difference between the total destruction and production rates. Red shaded bands indicate the 1σ uncertainty due to experimental errors of the measured quantities (Table S1) and the reaction rate coefficients. The pink shaded areas represent the maximum possible bias from a potential OH interference.

**Anonymous Referee #3**

*Comments*

(1)

*page 1, line 19: "In case of RO2, the budget can only be closed when the missing OH reactivity is attributed to unmeasured VOCs. Thus, the existence of unmeasured VOCs is directly confirmed by RO2 measurements." This is a likely but not exclusive explanation. I recommend to rephrase the sentences: "In case of RO2, the budget could be closed by attributing the missing OH reactivity to unmeasured VOCs. Thus, unmeasured VOCs are directly linked to the RO2 measurements."*

We changed the sentence as follows.
In case of $RO_2$, the budget could be closed by attributing the missing OH reactivity to unmeasured VOCs. Thus, the presumption of the existence of unmeasured VOCs is supported by $RO_2$ measurements.

(2)

*page 1, line 25: "These observations suggest the existence of a chemical mechanism that converts RO2 to OH without the involvement of NO. Please quantify the average contribution of this channel to the total turnover rate.*

We added conversion rates in the sentence, which reads now
These observations suggest the existence of a chemical mechanism that converts $RO_2$ to OH without the involvement of NO, increasing the $RO_2$ loss rate at daytime from 5.3 ppbv/h to 7.4 ppbv/h, on average.

(3)

*page 2, line 36: " ... tendency to underpredict OH under low-NOx conditions" . Please define the term "low-NOx conditions".*

We changed the sentence to
... tendency to underpredict OH under low-NOx conditions (NO < 300 pptv).

(4)

*Page 4, line 4: Please explain in the supplement table caption the term "RO2#".*

In the footnotes of Table S1, we included the following explanation.
$RO_2$# are organic peroxy radicals from large alkanes (> C4), alkenes (including isoprene) and aromatics.

(5)

*Page 4, line 12: Please quantify how much OH is internally removed.*

We added the following the sentence.
In the OH detection cell, scavenging of artificially produced OH by the added propane is calculated to be less than 0.3%.

(6)

*Page 4, line 13: How did you account for possible impurities in N2?*

We added the following explanation.
For nitrogen, we used research grade purity (>99.9990%). A GC analysis of the nitrogen showed no significant contamination by VOCs, which would scavenge OH in the chemical modulation system.

(7)

*Page 4, line 21: Please quantify the upper limit of the HO2 interference.*

We added the value (5%) for the upper limit of the interference to the text.

(8)

*Page 4, line 33: How did you quantify the HO2 background signal?*

We added the following explanation on Page 4 Line 33.
The signal was regularly determined in humidified synthetic air during calibration and found to be stable over the campaign. It is equivalent to $(2\pm1)\times10^7\,cm^{-3}$ and $(1\pm1)\times10^7\,cm^{-3}$ for $HO_2$ and $RO_2$, respectively, and is routinely subtracted from the measurements.

(9)

*Page 5, line 8: It is unclear which instruments have been used for CO and CO2 in this study.*

We added a sentence on Page 5 Line 9.
The CO and $CO_2$ measurements from the Thermo Electron and Picarro instruments agreed within the instrumental accuracies. The Picarro measurements were used in this work due to the better data coverage.

(10)

*Page 6, line 36: Please quantify the impact of the assumption that the nitrate yield is 5%. Please add a reference for the nitrate yield.*

In the text (page 6, line 36), we refer to section 2.3.1 where we have provided references and the range of possible nitrate yields. The impact of our assumption of a yield of 5% is explained in the discussion (Section 4.2, Additional uncertainties in the budget analyses) and in the corresponding Supplementary Text).

(11)

*Page 7, line 17: Despite the fast NO3 photolysis Liebmann et al. , 2018, found during daytime a fractional loss of NO3 of 25% by reaction with BVOC. What is the daytime NO3 production rate in this study and what would be an upper limit for its contribution?*

This is a very good question. We revised the text as follows.

NO₃ is produced by reaction of $NO_2$ with ozone. It is generally assumed that during the bright hours of the day, $NO_3$ is predominantly destroyed by photolysis and reaction with NO. Recently, Liebmann et al. (2018) reported measurements in a forested environment in southern Germany demonstrating that more than 25% of daytime $NO_3$ was removed by biogenic VOCs. The possible role of $NO_3$ reactions with VOCs at Heshan is discussed in Section 3.4 and 4.2.

In the presentation of the ROx budget (Section 3.4) a comment about $NO_3$ was included (see Comment 3 to the editor). In the general discussion of uncertainties of the radical budgets (Section 4.2) the following text was added.

Reactions of $NO_3$ with VOCs are an additional $RO_2$ source which is neglected in the budget calculation in Section 2.3.4. The relevance can be estimated from the production rate of $NO_3$, which is calculated from the reaction of $NO_2$ with $O_3$ ($k(NO_2+O_3)= 1.47\times10^{-13}\times exp(-2470/T)$; MCM3.3.1). In this campaign, the $NO_3$ production rate was in the order of 1.4 ppbv/h and 0.7 ppbv/h at day- and nighttime, respectively. Because $NO_3$ is efficiently photolysed in the bright hours of the day, it is generally assumed that it plays a negligible role as an oxidant during daytime. Liebmann et al. (2018) have recently shown that this is not always the case. They reported measurements in a forested environment in southern Germany demonstrating that more than 25% of the daytime $NO_3$ reacted with biogenic VOCs. Under the conditions at Heshan 2014, the main loss process at daytime is the reaction with NO. If we neglect unmeasured VOCs, the percentage removal of $NO_3$ in the morning is 96% by NO, 3% by photolysis, and 1% by measured VOCs. In the afternoon, the corresponding values are 72%, 21%, and 7%. Thus, the estimated $RO_2$ production rate from $NO_3$ reactions with known VOCs was probably not more than 0.1 ppbv/h at daytime. It is conceivable, that unmeasured VOCs, which probably accounted for 50% of the OH reactivity, contributed by a similar magnitude. During daytime, these contributions are relatively small compared to the total production rate of $RO_2$. The tendency is to slightly increase the imbalance between the production and destruction rate of $RO_2$ observed in the afternoon (Fig. 3). At sunset and in the night, the $NO_3$ production rate of 0.7 ppbv/h can be considered as an upper limit for the $RO_2$ production. This value can possibly explain at least partly the imbalance of about 0.5 ppbv/h in the ROx budget after sunset (Fig. 2).

(12)

*Page 9, line 38, What is the upper limit of ROx production by NO3 during night time, i.e. what is the NO3 production rate?*

See our reply to comment (11) above.

(13)

*Page 10, line 24 Is it the only exclusive explanation or a possible explanation that fits the result ?*

It is a possible and plausible explanation. We revised the sentence.

Once the missing OH reactivity is attributed to unmeasured VOCs, the resulting production rate $P^{(2)}_{RO2}$ calculated by Equation E8 (Fig. 3e) matches $D_{RO2}$ relatively well.

(14)

*Page 11, line 6: "The completeness of the radical measurements allows a budget analysis for all radicals (OH, HO2, RO2) based on experimental data only, ...". Please add: "under the assumption that for the production and loss rates all relevant species were measured."*

We added the sentence as suggested.

(15)

*Page 11, line 9: How do you define daytime?*

In Section 3.1 we added the following definition.
In this work, conditions with $j_{O1D} > 1{\times}10^{-6}$ s$^{-1}$ are referred to as daytime conditions lasting from 6:00 to 18:00 local time.
On page 11, line 9, we added the time window (6:00 to 18:00).

(16)

*Page 11, line 30: It is not obvious that under NOx regimes, controlling radical propagation and termination schemes, the resulting intermediates or even the emitted VOC found in Yang et al are comparable with the ones in this study.*

We agree and changed the sentence.
Although the percentage value is smaller than in the present paper (50%), the absolute values for the OH reactivity from unmeasured reactants are comparable. The speciation of the missing reactivity, however, can be different because the higher NOx loading in the period analyzed by Yang et al. (2017) may lead to different photochemical products and may be correlated with different VOC emissions.

(17)

*Page 12, line 4: Please specify uncertainties.*

We modified the sentence and added uncertainties.
The imbalances in the OH, HO$_2$, and RO$_2$ budgets (*D-P*) reach median values of up to (7±2.5) ppbv/h, -(3±5) ppbv/h, and -(5±2.5) ppbv/h, respectively, during the day.

(18)

*Page 13, line 35 Please quantify "negligible" including upper limit for dry deposition*

We modified the sentence and added loss rates for ozone.
Chemical loss of ozone by photolysis (R2), ozonolysis reactions (R4) and dry deposition is neligible under the given conditions. Calculated losses of ozone by photolysis and ozonolysis are not larger than 0.2 ppbv/h. The dry deposition rate at daytime is estimated to be no more than 1 ppbv/h assuming a mixed boundary layer height of 1km and a maximum deposition velocity of 1 cm/s (e.g., Weseley et al., 2000).

(19)

*Page 13, line 36&37; Page 14 line 7 : Specify uncertainties*

We added uncertainties. The sentence reads now

… a daily integrated net ozone production of (102±31) ppbv is calculated (06:00 h to 18:00h). For comparison, the daily integrated OH+$NO_2$ term is (14±3) ppbv.

(20)

*Page 14, line 4: Please include the statement that the loss term HO2\*NO generating NO2 can be replaced by the production term of HO2 under the assumption that other HO2 losses, like HO2+RO2, HO2+HO2 are negligible.*

We added the following sentence.

[revised manuscript text omitted]
. Information about the abundance of alkenes in this campaign can be obtained from the RO$_2$# budget analysis. RO$_2$# is produced by OH reaction with alkenes, aromatics and large alkanes. The budget analysis (Fig. 3) shows that the calculated production rate $P^{(1)}_{RO2}$ of RO$_2$# from these compounds is balanced by the calculated RO$_2$# loss rate. If an essential fraction of the unmeasured VOCs would consist of alkenes, it would increase the RO$_2$# production rate correspondingly. Within experimental uncertainty, a doubling of the alkene contribution in the RO$_2$# production would be acceptable without disturbing the balance in the RO$_2$# budget. Doubling of the alkenes would explain 15% of the missing OH reactivity. In this case, the radical production from ozonolysis, which is less than 0.1 ppbv/h for OH and 0.05 ppbv/h for HO$_2$ at daytime, would increase by about a factor of 2. This increase would have a negligible impact on the radical budgets of OH and HO$_2$.  Unmeasured OVOCs could form additional radicals (HO$_2$, RO$_2$)

through photolysis. Such reactions would further increase the gap between the production and destruction rate for $RO_2$ and disturb the closed $RO_x$ and $HO_2$ budgets.

*Radical initiation by Cl atoms*

Gaseous nitryl chloride ($ClNO_2$) can be formed at night by heterogeneous reaction of $N_2O_5$ with chloride in moist particles (e.g., Osthoff et al., 2008). In the morning, $ClNO_2$ photolyzes and forms Cl atoms which react very fast with VOCs and produce additional $RO_2$. This mechanism can play a role for 2 - 3 hours after sunrise until the $ClNO_2$ reservoir is depleted. $ClNO_2$ was not measured in Heshan, but was reported for other places in China. Measured concentrations shortly before sunrise are typically below 1 ppbv (e.g., Tham et al., 2016; Wang et al. (2018)), but can occasionally reach a few ppb (e.g., 2.1 ppbv in Wangdu, Tham et al. (2016); 4.7 ppbv in Hong Kong,(Wang et al., 2016)). With photolytical lifetimes of 2 - 3 hours, Cl production rates rarely exceed 0.5 ppbv/h. $RO_2$ production with a similar rate will make only a minor contribution to the $RO_2$ budget (Fig. 3e), and make the balance in the ROx budget slightly worse (Fig. 2g).

*Uncertainties related to the measurement and chemistry of $RO_2$*

Uncertainties in the radical budgets may be caused by the measurement and incomplete representation of the $RO_2$ chemistry. Due to the measurement principle of the applied ROxLIF technique, only those $RO_2$ species can be measured which are converted to $HO_2$ by reaction with NO. This measurement is exactly what is needed to quantify the $HO_2$ production rate (equation E5) in the atmospheric $HO_2$ budget. However, using the measured $RO_2$ data for the calculation of the $RO_2$ loss rate (equation E9) may cause a systematic bias. There exist $RO_2$ radical species which react with NO and produce a new $RO_2$ radical rather than $HO_2$. An example is the reaction $(CH_3)_3C(O_2)+NO$ leading to $CH_3O_2$+acetone+$NO_2$ as products. The result is a low-biased measurement of atmospheric $RO_2$ radicals. Its use in equation E9 leads to an underestimation of $D_{RO2}$ since the $RO_2$ loss leading to new $RO_2$ species is not included due to the measurement bias. On the other side, the production $P_{RO2}$ in equation E8 is also underestimated by the same amount, because the production term for $RO_2$ species which are produced by $RO_2$+NO is missing. As a result, the balance term $D_{RO2}$-$P_{RO2}$ in Fig. 2 remains correct as the production and destruction terms are smaller by the same unknown amount. Another group of $RO_2$ radicals which are not well captured by ROxLIF are nitrate peroxy radicals which are formed by the reaction of $NO_3$ radicals with alkenes. Some nitrate peroxy radical species (e.g., from propene and butenes) react with NO and produce besides $HO_2$ in a parallel reaction carbonyl compounds and $NO_2$ as products. The latter reaction constitutes a ROx sink. In the present work, $NO_3$ reactions with VOCs play a minor role (Section 4.2).

Other uncertainties in the $RO_2$ budget are caused by the rate constants that are given in Table 1 as effective values for the lumped $RO_2$ radicals. It is well known that the rate coefficients for the reactions of $RO_2$ with NO, $HO_2$, and $RO_2$ depend on the chemical structure of the $RO_2$ species. According to Jenkin et al. (2019), experimentally known rate constants for $RO_2$+NO can be broadly categorized into three classes: [1] $CH_3O_2$ (C1), [2] other hydrocarbon ($\geq$ C2) and oxygenated peroxy radicals, and [3] acyl peroxy radicals. At room temperature,

recommended rate constants for these categories are $7.7 \times 10^{-12}$ cm$^3$s$^{-1}$, $9.0 \times 10^{-12}$ cm$^3$s$^{-1}$, and $2.0 \times 10^{-11}$ cm$^3$s$^{-1}$, respectively (Jenkin et al., 2019). The MCM value used in Table 1 for R8 + R14 ($9.0 \times 10^{-12}$ cm$^3$s$^{-1}$) fits to the second class. The high rate constants for acyl peroxy radicals have no relevance for the budget analysis, because their reaction with NO produces another RO$_2$ radical. Thus, their reaction does not contribute to the HO$_2$ production and is neutral in the RO$_2$ budget as explained above. Published rate constants of the second category range between $8 \times 10^{-12}$ cm$^3$s$^{-1}$ and $1.1 \times 10^{-11}$ cm$^3$s$^{-1}$ (Jenkin et al., 2019). Here, the lower limit is almost equal to the rate coefficient of CH$_3$O$_2$ (first class). As a sensitivity test, Figs. S5 and S6 show the budgets of ROx, RO$_2$ and HO$_2$ for a rate constant of $1 \times 10^{-11}$ cm$^3$s$^{-1}$ (R8 + R14). The results are essentially the same as in Figs. 2 and 3 where a rate constant of $9 \times 10^{-12}$ cm$^3$s$^{-1}$ was applied. As the RO$_2$ budget indicates a missing RO$_2$ sink, a larger rate constant could help resolve the discrepancy. However, the 10% increase of the rate constant for R8 + R14 in Figs. S5 and S6 is far too small to explain the observed imbalance.

The reaction of RO$_2$ radicals with NO can form HO$_2$ (reaction R8) resulting in radical chain propagation, or produce organic nitrates (reaction R14) resulting in chain termination. As the branching ratio can be different for each RO$_2$ species and as most of the organic reactivity was caused by unmeasured VOCs, the branching ratios of most RO$_2$ species are not known. Typical yields for organic nitrates lie in the range between 1% and 35% (Atkinson (Atkinson et al., 1982)., 1982; Lightfoot et al., 1992). For the budget analysis (Figs. 2 - 4), an organic nitrate yield of 5% is assumed. Figs. S4 S7 and S5 S8 show cases where higher yields (10%, 20%) are assumed. Higher organic nitrate yields compensate the slightly negative bias of $D$-$P$ in the RO$_x$ budget (Fig. S4S7). An average yield of 10% would lead to a perfect balance between production and destruction rate of ROx during daytime, whereas a yield of 20% would result in a slightly positive bias of up to +1 ppbv/h in $D$-$P$. For the HO$_2$ production rate, these changes have little impact. Thus, in all cases (80%, 90%, 95% yield of HO$_2$), the HO$_2$ budget is balanced within the experimental uncertainties.

Published rate constants for the reaction RO$_2$+HO$_2$ (R16) lie in the range between $0.5 \times 10^{-11}$ cm$^3$s$^{-1}$ and $2.2 \times 10^{-11}$ cm$^3$s$^{-1}$ at 298K (Jenkin et al., 2019). In MCM, a general value of $2.3 \times 10^{-11}$ cm$^3$s$^{-1}$ (298K) is assumed and scaled by an RO$_2$ specific factor which is typically 0.5 - 0.7. In the budget analysis we have used the upper limit with a scaling factor of one. Thus, the possible bias of the calculated RO$_2$+HO$_2$ rate is in the order of a factor of 2. Under the polluted conditions of the campaign, the loss of RO$_2$ and HO$_2$ is largely dominated by NO. The reaction RO$_2$+HO$_2$ contributes only a few percent to the ROx loss during daytime and no more than 10% at sunset, when NO is small. Thus, the bias in the calculated ROx loss rate remains well below 5% at daytime. Similar considerations apply to the loss of RO$_2$ and HO$_2$, which is also dominated by NO during the day.

Rate coefficients for self and cross reactions of RO$_2$ are diverse and difficult to parameterize (Jenkin et al., 2019). The rate constants for the most abundant species are generally an order of magnitude smaller than for the reaction R16 (RO$_2$+HO$_2$). Self reactions of oxygenated RO$_2$ and cross reactions of some RO$_2$ can be as fast as reaction R16 (Jenkin et al., 2019). Overall, RO$_2$+RO$_2$ reactions play a smaller role than RO$_2$+HO$_2$ reactions in the Heshan campaign. The uncertainty of the RO$_2$ radical budget due to the lumped rate coefficient for R15 is therefore negligible.

**Table S1 Measured quantities used to evaluate the radical budgets.**

| Measured quantity | Measurement technique | Time resolution | Detection limit [a] | Accuracy (1σ) |
|---|---|---|---|---|
| OH | LIF [b] | 300 s | $3.9 \times 10^5 \, cm^{-3}$ | ±13 % |
| $HO_2$ | LIF [b, c] | 300 s | $1.2 \times 10^7 \, cm^{-3}$ | ±20 % |
| $RO_2$ | LIF [b, c] | 300 s | $0.6 \times 10^7 \, cm^{-3}$ | ±26 % |
| $RO_2^{\#}$ [d] | LIF [b, c] | 300 s | $1.7 \times 10^7 \, cm^{-3}$ | ±32 % |
| $k_{OH}$ | LP-LIF [e] | 180 s | $0.3 \, s^{-1}$ | ±10 %, ±0.7 $s^{-1}$ |
| Photolysis frequencies | Actinic flux spectroradiometry | 20 s | f | ±10 % |
| $O_3$ | UV photometry | 60 s | 0.5 ppbv | ±5 % |
| NO | Chemiluminescence | 60 s | 60 pptv | ±20 % |
| $NO_2$ | Chemiluminescence [g] | 60 s | 300 pptv | ±20 % |
| HONO | LOPAP [h] | 30 s | 7 pptv | ±20 % |
| $CO, CH_4, CO_2, H_2O$ | Cavity ringdown spectroscopy | 60 s | i | j |
| $SO_2$ | Pulsed UV fluorescence | 60 s | 0.1 ppbv | ±5 % |
| HCHO | Hantzsch fluorimetry | 60 s | 25 pptv | ±5 % |
| NMHCs [k] | GC-FID/MS [l] | 1 h | 20 - 300 pptv | ±(15-20) % |

[a] Signal to noise ratio = 1; [b] Laser-induced fluorescence; [c] Chemical conversion via NO reaction before detection; [d] $RO_2^{\#}$  are the organic peroxy radicals  from large alkanes (> C4), alkenes (including isoprene) and aromatics; [e] Laser photolysis – laser-induced fluorescence; [f] Five orders of magnitude lower than maximum at noon; [g] Photolytic conversion to NO before detection, home built converter; [h] Long-path absorption photometry; [i] CO: 1 ppbv; $CH_4$:1 ppbv; $CO_2$: 25 ppbv; $H_2O$: 0.1 % (absolute water vapor content).; [j] CO: ±1 ppbv; $CH_4$: ±1 ppbv; $CO_2$: ±25 ppbv; $H_2O$: ±5 % ; [k] NMCHs include $C_2$-$C_{11}$ alkanes, $C_2$-$C_6$ alkenes, $C_6$-$C_{10}$ aromatics; [l] Gas chromatography equipped with mass spectrometer and a flame ionization detector.

**Table S2 Measured volatile organic compounds.**

| Groups | VOC compounds |
|---|---|
| **Alkanes** | CYCLOHEXANE, CYCLOPENTANE, ETHANE, I-BUTANE, I-PENTANE, METHYLCYCLOHEXANE, METHYLCYCLOPENTANE, N-BUTANE, N-DECANE, N-DODECANE, N-HEPTANE, N-HEXANE, N-NONANE, N-OCTANE, N-PENTANE, N-UNDECANE, PROPANE, 2,2,4-TRIMETHYLPENTANE, 2,2-DIMETHYLBUTANE, 2,3,4-TRIMETHYLPENTANE, 2,3-DIMETHYLBUTANE, 2,3-DIMETHYLPENTANE, 2,4-DIMETHYLPENTANE, 2-METHYLHEPTANE, 2-METHYLHEXANE, 2-METHYLPENTANE, 3-METHYLHEPTANE, 3-METHYLHEXANE, 3-METHYLPENTANE |
| **Alkenes** | CIS-2-PENTENE, CIS-BUTENE, ETHENE, I-BUTENE, PROPENE, TRANS-2-BUTENE, TRANS-2-PENTENE, 1-BUTENE, 1-HEXENE, 1-PENTENE, STYRENE [a] |
| **Aromatics** | BENZENE, ETHYLBENZENE, I-PROPYLBENZENE, M-DIETHYLBENZENE, M-ETHYLTOLUENE, M,P-XYLENE, N-PROPYLBENZENE, O-ETHYLTOLUENE, O-XYLENE, P-DIETHYLBENZENE, P-ETHYLTOLUENE, TOLUENE, 1,2,3-TRIMETHYLBENZENE, 1,2,4-TRIMETHYLBENZENE, 1,3,5-TRIMETHYLBENZENE |
| **Alkynes** | ETHYNE |
| **Biogenics** | ISOPRENE |
| **OVOCs** | FORMALDEYHYDE |

[a] Styrene is treated as alkene because its major  functional group is the C-C double bond with respect to OH reaction.

[Figure]

**Figure S1 Time series of measured photolysis frequencies, O₃, Oₓ (O₃+NO₂), NO, NO₂, HONO, CO, isoprene, styrene, HCHO, and H₂O volume mixing ratios, PM₂.₅ mass concentrations and surface area of particulate matter. The vertical dashed lines represent midnight and grey areas represent nighttime.**

[Figure]

**Figure S2 Time series of measured OH, HO₂, RO₂ and RO₂# concentrations. The lowest panel shows the measured total OH reactivity (*k*OH) and the calculated OH reactivity (*k*calcOH) derived from measured concentrations of CO, NOx, CH₄, NMHCs and HCHO. The vertical dashed lines represent midnight and grey areas represent nighttime.**

[Figure]

**Figure S3 Results from the chemical modulation tests performed on 31 October 2014 between 12:50 and 13:50. The measured OH signal without scavenger ($S_{N2}$) can be explained within experimental errors by the sum of the signal from ambient OH ($S_{OH}$) and the known interference from $O_3$ ($S_{O3}$). Error bars denote $1\sigma$ statistical errors. $S_{OH}$ is calculated by the expression ($S_{N2}$ - $S_{propane}$)/$\varepsilon$, where $S_{propane}$ is the signal with scavenger (propane) and $\varepsilon$ is the efficiency of scavenging (for details, see Tan et al., 2017). A fluorescence signal of 60 cts/s is equivalent to an OH concentration of $1\times 10^7$ cm$^{-3}$.**

[Figure]

**Figure S4 Same as Fig. 3, but with additional RO₂ conversion to OH assuming a first-order rate coefficient of 0.08 s⁻¹. This scenario can also be seen as an application of the X mechanism which recycles OH by the hypothetical sequence RO₂ + X → HO₂, HO₂ + X → HO₂ with X equivalent to 0.4 ppbv NO.**

[Figure]

**Figure S5 Same as Fig. 2, but assuming a rate constant of $1\times10^{-11}\,\mathrm{cm}^{-3}\mathrm{s}^{-1}$ for the reaction of $RO_2$ with NO (R8, R14) .**

[Figure]

**Figure S6 Same as Fig 3. (c, d, e, f) but assuming a rate constant of $1 \times 10^{-11} cm^{-3} s^{-1}$ for the reaction of RO$_2$ with NO (R8, R14) .**

[Figure]

**Figure S4 S7** Same as Fig. 2, but assuming a different branching ratio between reaction R8 and R14. Left: HO₂ yield is 0.8, organic nitrate yield is 0.2. Right: HO₂ yield is 0.9, organic nitrate yield is 0.1.

[Figure]

**Figure S5 S8** Same as Fig. 3(c, d), but assuming a different branching ratio between reaction R8 and R14. Left: HO$_2$ yield is 0.8, organic nitrate yield is 0.2. Right: HO$_2$ yield is 0.9, organic nitrate yield is 0.1.